**Changing global cropping patterns to minimize national blue water scarcity**
Hatem Chouchane[a]*, Maarten S. Krol[a], and Arjen Y. Hoekstra[a, b, †]
[a] Twente Water Centre, University of Twente, Enschede, The Netherlands
[b] Institute of Water Policy, Lee Kuan Yew School of Public Policy, National University of Singapore, Singapore
[*] Contact author: hatemchouchane1@gmail.com, Phone: + 31 53 489 4446
[†] Deceased 18 November 2019.
**Postal address**
University of Twente
Faculty of Engineering Technology
Civil Engineering
Department of Water Engineering & Management
P.O. Box 217
7500 AE Enschede
**Abstract**
Feeding a growing population with global natural resource constraints becomes an increasingly challenging task.
Changing spatial cropping patterns and international crop trade could contribute to sustain crop production and mitigate water
scarcity. Previous studies on water saving through international food trade focussed either on comparing water productivities
among food-trading countries or on analysing food trade in relation to national water endowments. Here, we consider, for the
first time, how both differences in water productivities and water endowments can be considered to analyse comparative
advantages of countries for different types of crop production. A linear optimization algorithm is used to find modifications in
global cropping patterns that reduce national blue water scarcity in the world's most severely water-scarce countries, under the
constraint of current global production per crop and current cropland areas. The optimization considers national water and land
endowments as well as water and land productivity per country per crop. The results are used to assess national comparative
advantages and disadvantages for different crops. When allowing a maximum expansion of harvested area per crop per country
of 10%, the blue water scarcity in the world's most water-scarce countries can be greatly reduced. In this case, we could
achieve a reduction of the current blue water footprint of crop production in the world of 21% and a decrease of the total global
harvested and irrigated areas of 2% and 10% respectively.

**Keywords:** global food supply; spatial crop distribution; water scarcity; comparative advantage; optimization














## Introduction

Water scarcity poses a major societal and economic risk (WEF, 2019) and threat to biodiversity and environmental sustainability (Vörösmarty et al., 2010). Population growth and climate change are expected to worsen the situation and impose more pressure on freshwater resources everywhere (Vörösmarty et al., 2000; Parry et al., 2004). Since water consumption already exceeds the maximum sustainable level in many parts of the world (Hoekstra et al., 2012) and population growth in water-scarce countries alone could enforce global international trade in staple crops to increase by a factor of 1.4 to18 towards 2050 (Chouchane et al., 2018) solutions are urgently needed for a more sustainable allocation of the world's limited freshwater resources (Hoekstra, 2014; Konar et al., 2016).

Considerable debate has arisen over the last few decades on the pathways to overcome the problem of water scarcity and its implications (Gleick, 2003), especially for agriculture, the largest consumer of freshwater, accounting for 92% of water consumption globally (Hoekstra and Mekonnen, 2012). A growing number of studies addresses the question of how to mitigate problems related to blue water scarcity (Wada et al., 2014; Kummu et al., 2016). Some proposed solutions focus on better water management in agriculture (Evans and Sadler, 2008), for instance improving irrigation efficiency and precision irrigation (Sadler et al., 2005; Greenwood et al., 2010), better agricultural practices like mulching and drip irrigation (Mukherjee et al., 2010; Chukalla et al., 2015; Nouri et al., 2019), improved irrigation scheduling (Jones, 2004) and enhancing water productivity (Bouman, 2007; Molden et al., 2010; Pereira et al., 2012). Other suggested solutions focus on changing diets (Vanham et al., 2013; Jalava et al., 2014; Gephart et al., 2016) and reducing food losses (Munesue et al., 2015; Jalava et al., 2016) to diminish water consumption. Yet another category of studies focusses on spatial cropping patterns (Davis et al., 2017a; Davis et al., 2017b) and the role of international trade in saving water and in bridging the gap between national water demand and supply in water-short countries (Chapagain et al., 2006; Hoekstra and Hung, 2005). The trade in 'embedded water' (also known as virtual water trade) is the hidden flow of water if food or other commodities are traded from one place to another (Allan, 1998). According to international trade theory, countries can profit from trade by focussing on the production and export of goods for which they have a comparative advantage. What precisely constitutes comparative advantage is still subject to debate. Whereas Ricardo's theory of comparative advantage says that a country can best focus on producing goods for which they have relatively high productivity, the Heckscher-Ohlin theory states that a country can best specialize in producing and exporting products that use production factors that are comparatively most abundant. When focussing on the role of water in trade, the first theory would consider relative water productivity (crop per drop), while the second theory would look at relative water abundance (Hoekstra, 2013). Part of the literature on water saving through international food trade has focussed on comparing water productivities among food-trading countries (Chapagain et al., 2006; Yang et al., 2006; Oki et al., 2017), while other studies have concentrated on analysing food trade in relation to water endowments (Yang et al., 2003; Oki and Kanae, 2004; Chouchane et al., 2018). In a study for China, Zhao et al., (2019), evaluated spatio-temporal differences in regional water, land and labour productivity of agricultural and non-agricultural sectors across Chinese provinces, and defined

comparative advantage on that basis. These comparative advantages were used to track the driving force of virtual water
regional trade. Their findings suggest that differences in land productivity were the main forces shaping the pattern of virtual
water flows across Chinese regions while neither labour nor water productivity had significant influence.

In the current study, we consider, for the first time, how both differences in water productivity and water endowment can

be considered to analyse comparative advantages of countries for different types of crop production. While doing so, we also
consider differences between countries in land productivities (crop yields) and land endowments (available cropland areas).

Studies on spatial allocation of crop production, given differences in land and water productivity and endowments, are

sparse, particularly large-scale studies. In local or regional studies that study best crop choices given land and water
constraints, the focus is generally to maximize food production or agricultural value, without the requirement of fulfilling
overall crop demand. Osama et al., (2017), for example, employ a linear optimization model for some regions in Egypt to
maximize the net annual return by changing the cropping pattern, given water and land constraints, and conclude that some
crops are to be expanded while others are to be reduced. Another example of a regional study is Ye et al. (2018), who used a
multi-objective optimization model, considering the trade-offs between economic benefits and environmental impact of water
use when changing the cropping pattern in a case study for Beijing.

In a study for the US, Davis et al. (2017b) investigated an alternative crop distribution that saves water and improves

productivity while maintaining crop diversity, protein production and income. The only global study on changing cropping
patterns in order to reduce water use, to our knowledge, is Davis et al., (2017a), who combine data on water use and
productivity for 14 major crops and show that changing the distribution of these crops across the currently cultivated lands in
the world could decrease blue water use by 12% and feed an additional 825 million people. However, the current study has a
number of differences with Davis et al., (20017a). First, we are only changing cropping patterns while maintaining the same
global production per crop whereas Davis et al. (2017a) aim for a higher caloric and protein production while reducing water
use; that also results in a different global consumption pattern, which hampers the identification of potential water saving
effects of just production shifts amongst countries. Second, we consider a larger number of crops (125 crops including
vegetables, fruits and pulses which were not considered in Davis et.al., (2017a) study).

Although it has been widely acknowledged that the spatial water scarcity pattern in the world can be explained by where

crops are grown and how much they are irrigated (Wada et al., 2011; Mekonnen and Hoekstra, 2016), it has not yet been
studied how differences between countries in water and land productivities and endowments can be used to derive comparative
advantages of countries for specific crops, and how a change in the global cropping pattern can reduce water scarcity in the
most water-scarce places. Here, we explore how we can stepwise minimize the highest national water scarcity in the world by
changing cropping patterns and the related blue water allocation to crops. The spatial resolution of the country level reflects the
coarse resolution at which FAO monitors and reports water stress in the SDG framework (FAO, 2018); subnational
heterogeneity in water scarcity, that is significant in countries like USA or China, is not covered at this resolution. With
cropping pattern we mean the allocation of crops to rainfed and irrigated land in all countries in the world, where both rainfed
and irrigated area of each crop in each country is allowed to expand up to a modest maximum rate (factor α), while respecting
the bounds of current total rainfed and total irrigated area per country as well as the global production per crop. For this
purpose, we develop and apply a linear programming optimization algorithm considering a number of constraints. First, total
rainfed and irrigated harvested areas in each country should not grow beyond their extent in the reference period 1996-2005.
Second, the harvested area per country per crop can only expand by a limited rate (which will be varied), both for the rainfed
and irrigated area. Third, global production of each crop must remain the same as in the reference period. The optimization
takes into account both factor endowments (blue water availability, rainfed land availability and irrigated land availability) in
each country and factor productivities (blue water productivity in irrigation, and land productivities in rainfed and irrigated
lands) for each crop in each country. In order to focus on aspects of natural resource endowment and productivity in relation to
water scarcity, other important aspect such as socio-economic or national food self-sufficiency goals were left out of
consideration.
**Methods and data**
We developed a linear optimization algorithm in MATLAB. In the optimization we allow the global cropping pattern to
change, that is to grow crops in different countries than in the reference situation. In the optimization, the cropping areas by
crop, country and production system are the independent variables, and the following constraints are considered. First, both
total rainfed and total irrigated harvested area per country are not allowed to expand. Second, both crop-specific rainfed and
irrigated harvested area per country are allowed to expand, but not beyond a factor $\alpha$ (whereby we stepwise increase $\alpha$ from
1.1 to 2.0 in a number of subsequent experiments). Third, global production of each crop should remain equal to the global
production of the crop in the reference situation. For any cropping pattern, the water scarcity in each country is computed, and
the country with the highest water scarcity identified. The objective of the optimization is to minimize this highest water
scarcity. The algorithm continuously tries to reduce the blue water scarcity in the countries with the highest blue water scarcity
while disallowing blue water scarcity in any country to increase. The algorithm will thus tend to reduce and equalize blue
water scarcity in the most water-scarce countries.
We considered 125 crops of the main crops groups (cereals, fibres, fruits, nuts, oil crops, pulses, roots, spices, stimulants,
sugar crops and vegetables; for an extensive list of crops used see (Chouchane et al., 2019)); optimization was performed using
the linear optimization routine from the Optimization Toolbox of MATLAB.
Given the cropping pattern, production is computed per country and crop, both for rainfed and irrigated lands based on
the harvested area and crop yields:
$\forall i,j: P_{rf}(i,j) = A_{rf}(i,j) \times Y_{rf}(i,j)$
$\forall i,j: P_{ir}(i,j) = A_{ir}(i,j) \times Y_{ir}(i,j)$
$\forall i,j: P(i,j) = P_{rf}(i,j) + P_{ir}(i,j)$

whereby $P_{rf}(i,j)$, $P_{ir}(i,j)$ and $P(i,j)$ are the rainfed, irrigated and total production in country $i$ of crop j; $A_{rf}(i,j)$ and $A_{ir}(i,j)$
the rainfed and irrigated harvested area in country i for crop j; and $Y_{rf}(i,j)$ and $Y_{ir}(i,j)$ the rainfed and irrigated crop yield in
country $i$ for crop $j$.
Blue water scarcity (BWS) is defined per country $i$ as the total blue water footprint divided by the blue water availability
in the country (Hoekstra et al., 2012). The blue water footprint (BWF) refers to the volume of consumptive freshwater use for
irrigation that comes from surface and groundwater. Blue water availability is taken from FAO (2015) and refers to the total
renewable (internal and external resources) which is the long-term average annual flow of rivers (surface water) and
sustainably available groundwater (FAO, 2003).

$BWS(i) = \dfrac{\sum_j P_{ir}(i,j) \times BWF(i,j)}{BWA(i)}$

where $P_{ir}(i,j)$ is the irrigated production in country $i$ of crop $j$, $BWF(i,j)$ the blue water footprint per unit of crop $j$ in country
$i$, and $BWA(i)$ the blue water availability in country $i$. A country is considered to be under low, moderate, significant or severe
water scarcity when BWS (expressed as a percentage) is lower than 20%, in the range 20-30%, in the range 30-40% and larger
than 40%, respectively (Hoekstra et al., 2012).
The optimization can be presented as follows:
$\min\limits_{A_{rf},A_{irr}} \left( \max\limits_{i}(BWS(i)) \right)$

subject to:
$\forall i: \sum\limits_j A_{rf}(i,j) \leq \sum\limits_j A_{rf,ref}(i,j)$
$\forall i: \sum\limits_j A_{ir}(i,j) \leq \sum\limits_j A_{ir,ref}(i,j)$
$\forall i,j: A_{rf}(i,j) \leq \alpha \times A_{rf,ref}(i,j)$
$\forall i,j: A_{ir}(i,j) \leq \alpha \times A_{ir,ref}(i,j)$
$\forall j: \sum\limits_i P(i,j) = \sum\limits_i P_{ref}(i,j)$
$\forall i : BWS(i) \leq BWS_{ref}(i)$

where $A_{rf}(i,j)$ and $A_{ir}(i,j)$ are the rainfed and irrigated harvested areas in country $i$ of crop $j$ in the cropping pattern that

is varied in order to minimize the highest national blue water scarcity, $A_{rf,ref}(i,j)$ and $A_{ir,ref}(i,j)$ are the rainfed and
irrigated harvested areas in the reference situation, $P(i,j)$ is the total (rainfed plus irrigated) production in country $i$ of crop $j$ in
the new cropping pattern, and $P_{ref}(i,j)$ is the total (rainfed plus irrigated) production in country $i$ of crop $j$ in the reference
situation, and $BWS_{ref}(i)$ is the blue water scarcity in country $i$ in the reference situation. Parameter $\alpha$ is the factor of
maximally allowed expansion of the harvested area per crop and country and production system (rainfed or irrigated), which is
varied in the optimization experiments between 1.1 and 2. Note that total national croplands (both rainfed and irrigated) are not
allowed to expand, but that reductions in land use are always allowed.

A country is considered to have a comparative advantage for producing a certain crop or crop group when the following

criteria are met: (1) the relative change (production in the optimized cropping pattern divided by the production in the
reference situation) of that crop or crop group continues to increase in that country when we gradually increase the maximum
allowed expansion of harvested area per crop per country (the factor $\alpha$); and (2) the share of the country in the global
production of the crop or crop group exceeds 5% (in the optimized cropping pattern at $\alpha = 1.1$).

In order to test the sensitivity of the optimization results to the allowed changes in irrigation, we run the optimization

without allowing any expansion of irrigated area. In this case, the factor $\alpha$ will be only applied to the rainfed area while the
irrigated area per country per crop will be below or equal to the irrigated area of the same crop in the same country in the
reference situation. The optimization objective and constraints remain the same except that the following constraint was added:
$\forall i,j : A_{ir}(i,j) \leq A_{ir,ref}(i,j)$

The sources of the data used to perform the optimization are summarized in Table 1.

**Table 1.** Overview of data used.

| Variable | Spatial resolution | Temporal resolution | Source |
|---|---|---|---|
| Blue water availability | Country (internal + external renewable water resources) | Average for 1961-1990 | (FAO, 2015) |
| Harvested irrigated and rainfed land per crop in the reference situation | Country | Average for 1996-2005 | (Mekonnen and Hoekstra, 2011), (FAO, 2015) |
| Rainfed and irrigated production per crop in the reference situation | Country | Average for 1996-2005 | (Mekonnen and Hoekstra, 2011), (FAO, 2015) |
| Blue WF per unit of crop in irrigated production per crop | Country | Average for 1996-2005 | (Mekonnen and Hoekstra, 2011) |
| Yield in rainfed and irrigated production per crop | Country | Average for 1996-2005 | (Mekonnen and Hoekstra, 2011) |

## Results

### Changes in blue water scarcity and blue water consumption

When α is 1.1, that means when we allow a maximum of 10% expansion of the reference harvested areas for each individual crop, in every country, both for rainfed and irrigated production, blue water scarcity in the world's seven most water-scarce countries, Libya, Saudi Arabia, Kuwait, Yemen, Qatar, Egypt, and Israel (with current scarcities ranging from 54% to 270%) is reduced to a scarcity of 39% or less (Table 2). In this scenario, the aggregated blue water footprint of crop production in the world is reduced by 21%, while the total global harvested and irrigated areas got reduced by 2% and 10% respectively.

When α is equal to 1.3, 1.5 and 2.0 (i.e., when the maximally allowed expansion of harvested area per crop per country is equal to 30%, 50% and 100%), the maximum national blue water scarcity in the world is further reduced to 6%, 4% and 2%, respectively. In these scenarios, global blue water consumption gets reduced by 38%, 48% and 60%, respectively, the total global harvested area gets reduced by 6%, 7% and 9%, respectively and the total global irrigated area gets reduced by 23%, 27% and 37% respectively.

**Table 2.** Current versus optimized blue water consumption (BWC) and blue water scarcity (BWS) for countries currently having a water scarcity higher than 15%.

| Countries | Current | | Optimized $(\alpha = 1.1)$ | | Optimized $(\alpha = 1.3)$ | | Optimized $(\alpha = 1.5)$ | | Optimized $(\alpha = 2.0)$ | |
|---|---|---|---|---|---|---|---|---|---|---|
| | BWC $(10^6 \, \text{m}^3/\text{yr})$ | BWS (%) | BWC $(10^6 \, \text{m}^3/\text{yr})$ | BWS (%) | BWC $(10^6 \, \text{m}^3/\text{yr})$ | BWS (%) | BWC $(10^6 \, \text{m}^3/\text{yr})$ | BWS (%) | BWC $(10^6 \, \text{m}^3/\text{yr})$ | BWS (%) |
| Libya | 1900 | 270% | 210 | 30% | 41 | 6% | 25 | 4% | 16 | 2% |
| Saudi Arabia | 6200 | 260% | 940 | 39% | 140 | 6% | 87 | 4% | 54 | 2% |
| Kuwait | 48 | 240% | 8 | 39% | 1 | 6% | 1 | 4% | 0 | 2% |
| Yemen | 2100 | 98% | 2.8 | 0% | 3 | 0% | 76 | 4% | 48 | 2% |
| Qatar | 51 | 88% | 23 | 39% | 3 | 6% | 2 | 4% | 1 | 2% |
| Egypt | 34000 | 57% | 3800 | 7% | 3400 | 6% | 2100 | 4% | 1300 | 2% |
| Israel | 960 | 54% | 340 | 19% | 100 | 6% | 65 | 4% | 40 | 2% |
| Jordan | 410 | 43% | 70 | 8% | 55 | 6% | 34 | 4% | 21 | 2% |
| Syria | 7000 | 42% | 690 | 4% | 990 | 6% | 610 | 4% | 380 | 2% |
| Oman | 550 | 39% | 550 | 39% | 82 | 6% | 51 | 4% | 32 | 2% |
| Uzbekistan | 15000 | 31% | 13000 | 26% | 890 | 2% | 1800 | 4% | 1100 | 2% |
| Cyprus | 240 | 31% | 59 | 8% | 46 | 6% | 28 | 4% | 18 | 2% |
| Pakistan | 74000 | 30% | 15000 | 6% | 14000 | 6% | 9000 | 4% | 5600 | 2% |
| Iran | 40000 | 29% | 8400 | 6% | 8000 | 6% | 5000 | 4% | 3100 | 2% |
| Tunisia | 1300 | 29% | 530 | 11% | 270 | 6% | 170 | 4% | 100 | 2% |

| Algeria | 2700 | 23% | 1900 | 16% | 690 | 6% | 430 | 4% | 260 | 2% |
|---|---|---|---|---|---|---|---|---|---|---|
| Turkmenistan | 5300 | 21% | 520 | 2% | 620 | 3% | 900 | 4% | 560 | 2% |
| Morocco | 5100 | 18% | 3100 | 11% | 1700 | 6% | 1100 | 4% | 660 | 2% |
| Malta | 9 | 17% | 8 | 15% | 3 | 6% | 2 | 4% | 1 | 2% |
| Lebanon | 770 | 17% | 730 | 16% | 260 | 6% | 160 | 4% | 100 | 2% |
| Sudan | 6100 | 16% | 2100 | 6% | 2200 | 6% | 1400 | 4% | 860 | 2% |
| **Global** | **820000** | | **650000** | | **510000** | | **430000** | | **330000** | |


Most countries with severe water scarcity (BWS>40%) in the reference situation show a moderate (BWS in the range 20-
30%) to low water scarcity (BWS<20%) in the optimized situation with α = 1.1 (Figure 1). However, not all countries would
benefit similarly in the optimized situation. China and India, major crops producers in the reference situation, only start to have
a decrease in their BWS when α ≥ 1.3.

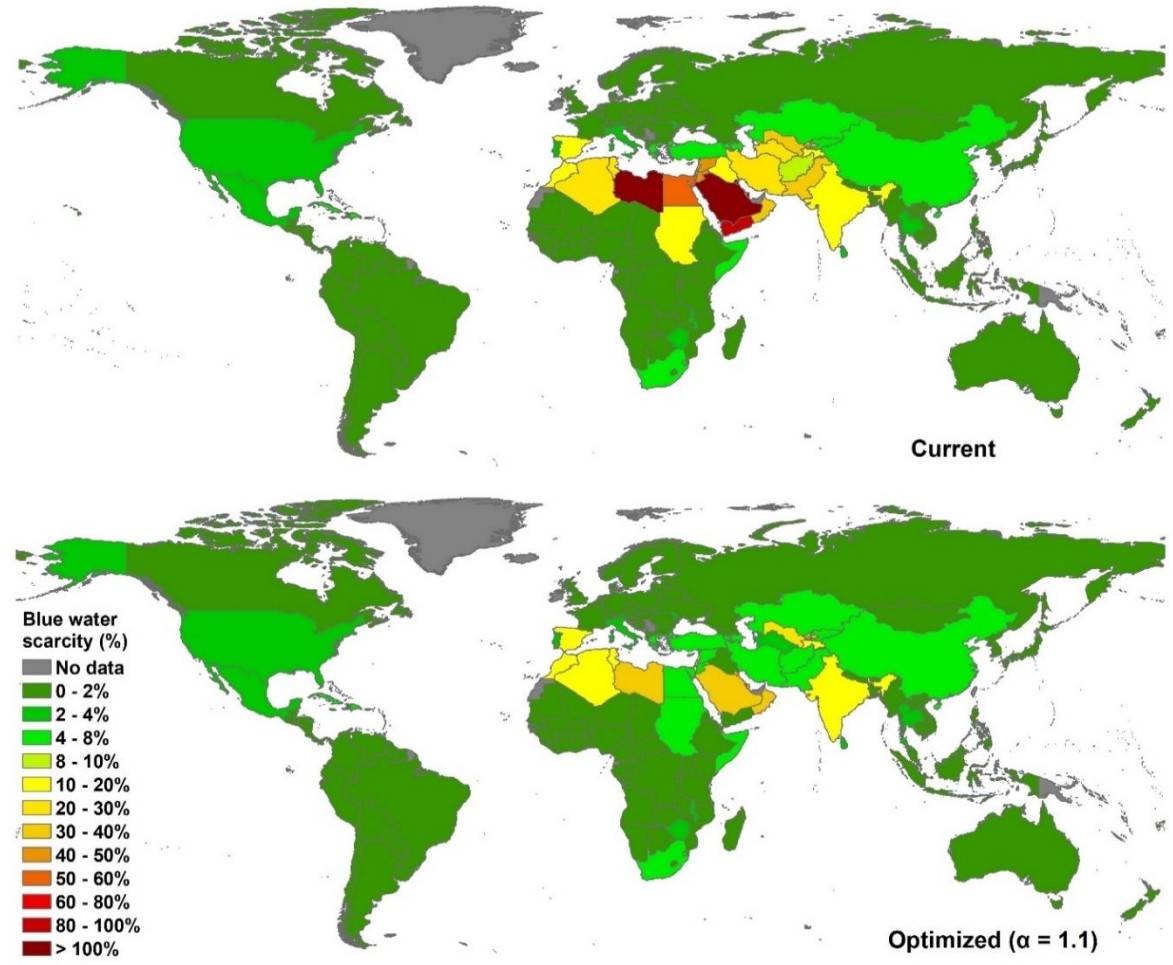

**Figure 1.** Current and optimized (α = 1.1) blue water scarcity.

In the case of α = 1.1, Pakistan, the 3$^{rd}$ largest consumer of blue water in the reference situation, has the largest reduction
in its blue water consumption in absolute terms, viz. 60,000 m$^3$/yr, which represents 80% of its current BWC and 35% of the
global blue water saving. Other countries that have a significant reduction in their BWC in absolute terms include Iran, Egypt,
Iraq, Syria, Saudi Arabia, Sudan and Turkmenistan (Figure 2). However, not all countries would benefit similarly in the
optimized set, India and China, the first and second largest consumer of blue water in the reference situation, will only start to
have a decrease in their blue water scarcity when the allowed expansion rate α is larger than 1.2; this is due to the optimization
of water scarcity at the level of countries, where India and China have modest national water scarcity.

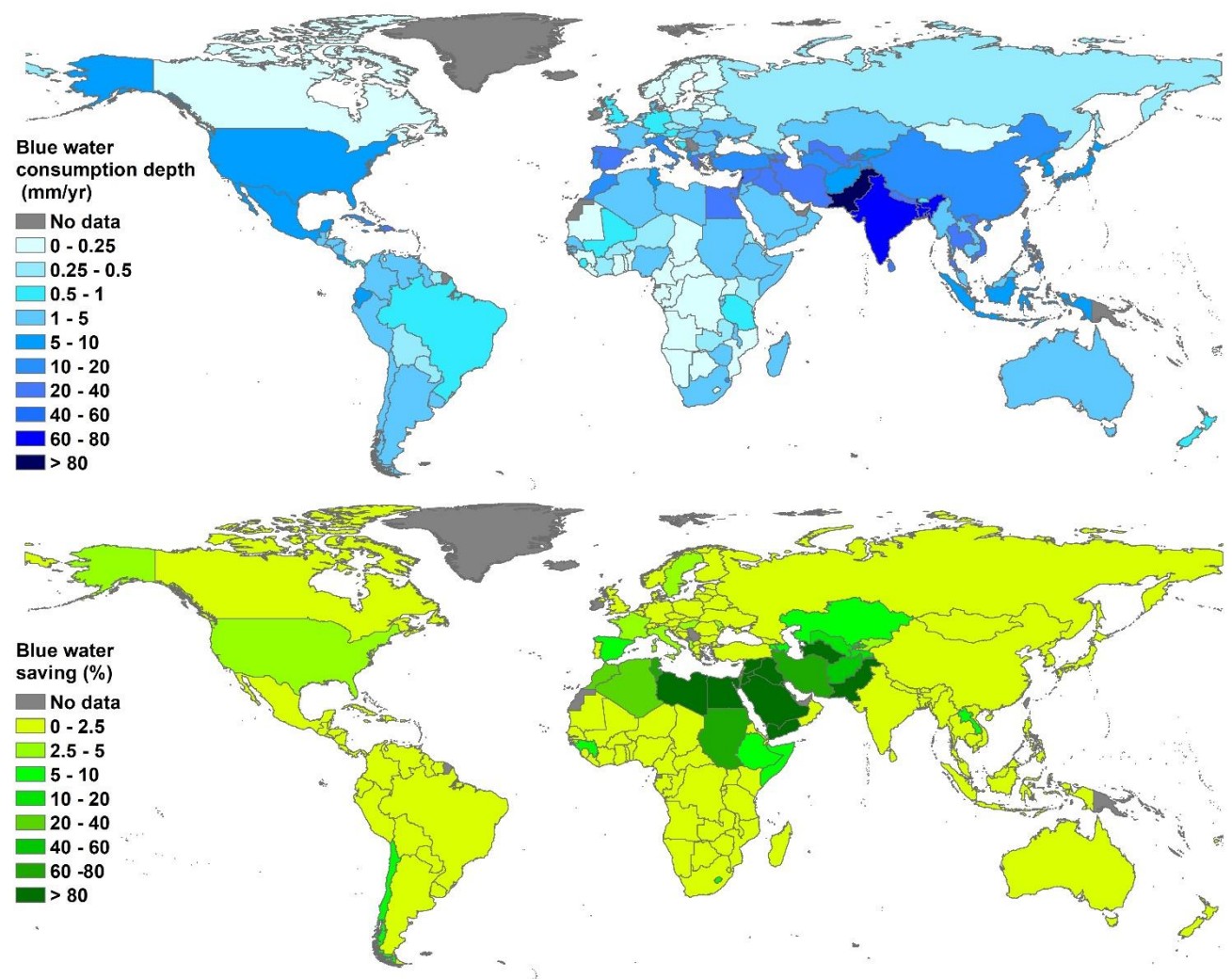


**Figure 2.** Current blue water consumption depth in mm/yr and blue water saving as a percentage of current BWC in the case of
an optimized cropping pattern (α = 1.1).
**The changing global cropping pattern for the case of α = 1.1**
The reduction of global blue water consumption is achieved by reallocating the most resource-intensive crops from
countries that have lower productivity in terms of land and water to countries with significantly higher productivities, both for
rainfed and irrigated production, and thus reducing irrigation in countries that initially have a high BWS. In the optimised
cropping pattern, cereal production is reduced most significantly in Africa, relative to the reference situation, and South
America and expanded in North America and Europe (Table 3). Irrigated cereal production is reduced in most world regions
(except for a small expansion in Europe and South America) whereas global rainfed production increases. For individual
countries, Pakistan and Egypt is the largest decrease in total cereal production. The most significant expansions in cereal
production are found in the US and China for Maize, in China, India, the Russian Federation and France for wheat production
and in India, Indonesia and Vietnam for rice production. In terms of harvested area, the largest areal decrease in cereals is
found in Asia with a reduction of 8 million hectares in total (Supplemental Table 1), which represent 3% of the current
harvested area of cereals in Asia. The irrigated area of cereals in Asia is reduced by 6% compared to the reference situation
while the rainfed area has an increase of 1%. Africa has the second-largest decrease of irrigated area of cereals with 3 million
hectares and the largest increase of rainfed area of cereals with 2.6 million hectares. Changes in the global pattern of cereal
production for the case of α = 1.1 contribute 50% to the total global reduction in the blue water footprint and 46% to the total
global reduction in irrigated area.
Fruit production is reduced most significantly in Asia and Africa and expanded in the Americas (Table 3). Major fruit
production reductions include the decrease of apple production in Iran, banana production in Thailand, orange production in
Egypt, Iran and Pakistan and grape production in France. In North America, the most significant expansion in fruit production
is the increase in orange, grape and apple production in the US; in South America, the largest fruit production increases are
oranges in Brazil and bananas in Ecuador. Although the reduction in fruit production in Asia and Africa mainly concerns
irrigation, the irrigated production of fruits increases in the North America and Europe. The largest share of increase in
irrigated production in North America comes from the increase in irrigated production of oranges, apples and grapes in the US.
The world's harvested area of fruits reduces by 2%. The irrigated area reduces by 19% while the rainfed area increases by 4%.
Changes in fruit production contributed 12% to global blue water savings and 9% to total global reductions in irrigated area.
**Table 3.** Change in production per product group per continent in absolute terms ($10^6$ t/yr) when shifting from the cropping
pattern in the reference period (1996-2005) to the optimized cropping pattern (with $\alpha = 1.1$)

| | | Cereal | Fibres | Fruits | Nuts | Oil crops | Pulses | Roots | Spices | Stimulants | Sugar crops | Vegetables |
|---|---|---|---|---|---|---|---|---|---|---|---|---|
| | Rainfed | 3.2 | 0.3 | 3.5 | 0.1 | -8.9 | 0.4 | 7.0 | 0.0 | 0.4 | 3.2 | 0.7 |
| Africa | Irrigated | -17.2 | -0.7 | -5.8 | 0.0 | -1.3 | -0.3 | -4.0 | -0.1 | 0.0 | -21.8 | -9.5 |
| | **Total** | **-14.0** | **-0.3** | **-2.3** | **0.1** | **-10.2** | **0.1** | **2.9** | **-0.1** | **0.4** | **-18.6** | **-8.9** |
| | Rainfed | 16.1 | 1.3 | 11.0 | 0.1 | 4.6 | -0.2 | 6.9 | 0.3 | 0.0 | 10.6 | 34.0 |
| Asia | Irrigated | -14.5 | -2.6 | -19.2 | -0.2 | -8.3 | -0.2 | -4.9 | -0.2 | -0.2 | -61.4 | -13.8 |
| | **Total** | **1.6** | **-1.3** | **-8.2** | **-0.1** | **-3.7** | **-0.4** | **1.9** | **0.1** | **-0.2** | **-50.8** | **20.1** |
| | Rainfed | 6.4 | 0.0 | -0.1 | 0.0 | 0.7 | -0.1 | -0.6 | 0.0 | 0.0 | 0.1 | -7.0 |
| Europe | Irrigated | 0.8 | 0.2 | 1.3 | 0.0 | 0.5 | 0.1 | 1.8 | 0.0 | 0.0 | 3.1 | -2.4 |
| | **Total** | **7.2** | **0.1** | **1.2** | **0.0** | **1.2** | **-0.1** | **1.3** | **0.0** | **0.0** | **3.3** | **-9.5** |
| | Rainfed | 11.6 | 0.6 | 1.2 | 0.0 | 5.1 | 0.5 | -0.9 | 0.0 | -0.2 | 8.9 | -1.0 |
| North America | Irrigated | -0.7 | 0.5 | 3.5 | 0.1 | 0.4 | 0.1 | 1.7 | 0.0 | 0.0 | 8.2 | -0.7 |
| | **Total** | **10.9** | **1.1** | **4.7** | **0.1** | **5.5** | **0.6** | **0.9** | **0.0** | **-0.2** | **17.1** | **-1.7** |

| | | | | | | | | | | | | |
|---|---|---|---|---|---|---|---|---|---|---|---|---|
| | Rainfed | 0.4 | 0.0 | 0.1 | 0.0 | 0.1 | -0.3 | -0.1 | 0.0 | 0.0 | 1.1 | -0.1 |
| Oceania | Irrigated | -0.3 | 0.1 | -0.1 | 0.0 | 0.0 | 0.0 | 0.1 | 0.0 | 0.0 | 2.9 | 0.1 |
| | **Total** | **0.1** | **0.1** | **-0.1** | **0.0** | **0.1** | **-0.3** | **0.1** | **0.0** | **0.0** | **4.0** | **0.0** |
| | Rainfed | -6.3 | 0.3 | 4.1 | 0.0 | 6.9 | 0.0 | -7.2 | 0.0 | 0.0 | 35.4 | -0.3 |
| South America | Irrigated | 0.6 | 0.0 | 0.6 | 0.0 | 0.1 | 0.0 | 0.2 | 0.0 | 0.0 | 9.6 | 0.3 |
| | **Total** | **-5.7** | **0.3** | **4.7** | **-0.1** | **7.0** | **0.1** | **-7.0** | **0.0** | **0.0** | **45.0** | **0.0** |

The production of oil crops is reduced most significantly in Africa (e.g. oil palm in Nigeria) and expanded in the Americas (e.g. soybeans in the US, Brazil and Argentina). The harvested area shrinks globally by 3% in total. Irrigated areas reduce by 30% although global rainfed area remain the same as the reference situation. Asia and Africa have the most significant shrinkage in harvested areas of oil crops. Reallocating oil crops contributed 7% to global reductions in blue water footprint and 19% to total global reductions in irrigated area.

Roots production partly moves from South America to Africa, Asia and Europe. At countries level, the most significant reduction is due to the decrease of potato production in Poland and Iran and cassava production in Brazil, China and Vietnam. The largest expansions are sweet potato production in China, potato production in the Russian Federation and Cassava and Yams in Nigeria. Globally, the harvested area of roots is reduced by 4% (11% for irrigated and 3% for rainfed croplands).

Sugar crop production is reduced most significantly in Asia and Africa and expanded in the Americas. Sugar cane production is mainly reduced in Pakistan, India and Egypt and expanded in Brazil. The global irrigated harvested area of sugar crops is reduced in total by 10% while the global rainfed area increases by 8% Changes in sugar crops production contribute 10% to the total blue water savings globally.

Vegetable production is reduced most significantly in Europe and Africa and expanded in Asia. Major reductions in vegetable production are for tomatoes production in Iran and Egypt. The most significant expansions are the increases in tomato and watermelon production in China. The global harvested area of vegetables is reduced by 4%, with a reduction of 17% for irrigated croplands while the rainfed area remains the same as reference situation. Reallocating vegetables contributed 5% to global reductions in blue water footprint and 7% to global reductions in total irrigated harvested area globally.

Although cereal rainfed harvested area is reduced in North America when $\alpha = 1.1$ for example (Supplemental Table 1), rainfed cereal production will increase by 11.6 million t/y. This illustrates that by allocating production to more productive countries we can reduce water and land use and increase production at the same time.

**Comparative advantages**

We explore comparative advantages of countries to contribute to the goal of relieving global water scarcity; in the following, we use the term "comparative advantage" to indicate comparative advantage for this specific goal, as that is where results from the study provide insight in; comparative advantages to e.g. contribute to increasing agroeconomic revenue or to

reduce agricultural carbon footprint could result in different conclusions. Our exploration of comparative advantage is
considering which crops in a country are expanding when we gradually move from α = 1.1 to α = 1.5. As a summary, Figure 3
shows at the level of continents and crop groups, the relative change in total production when we move from the reference
cropping pattern (period 1996-2005) along the optimized cropping pattern, considering a stepwise increase in the maximally
allowed expansion rate in harvested area per crop per country from $\alpha = 1.1$ to α = 1.5. Most of the changes in production
already occur for the modest areal expansion rate per crop of 10% (Table 3) will continue under larger expansion rates, with
some exceptions. This is, for example, the case for fibres in Europe and oil crops in North America. Fibres production expands
for the case of $\alpha = 1.1, 1.2$ and 1.3 in Europe but again reduces for higher expansion rates. This can be explained by the
fact that even more suitable regions, namely Oceania, North America and to a lesser extent Africa, continue expanding fibres
production,5 allowing Europe to rather focus on cereals, sugar crops and stimulants production (Figure 3). North America
expands oil crops production when $\alpha = 1.1$ (Table 3) but decreases oil crops production when $\alpha = 1.2$ and has the largest
reduction in oil crops production for $\alpha = 1.5$ (Supplemental Table 1). The reason behind this is that for the smallest expansion
rate, the US still needs to produce oil crops, and the global production could not be reached without the expansion of oil crops
in the US which limits the allocation of harvested areas to more suitable crops in the US such as maize and sugar crops. From
$\alpha = 1.2$ the US will focus on producing maize in which they have a comparative advantage and give up a part of oil crops
production. This example for North America shows that it is hard to have a robust conclusion on comparative advantages by
looking at the level of continents. In order to explore comparative advantages, we will need to look at country level. Figures 4
and 5 show the absolute and relative changes in production per crop group per country when we move from the cropping
pattern in the reference situation to the optimized cropping pattern with α = 1.5.

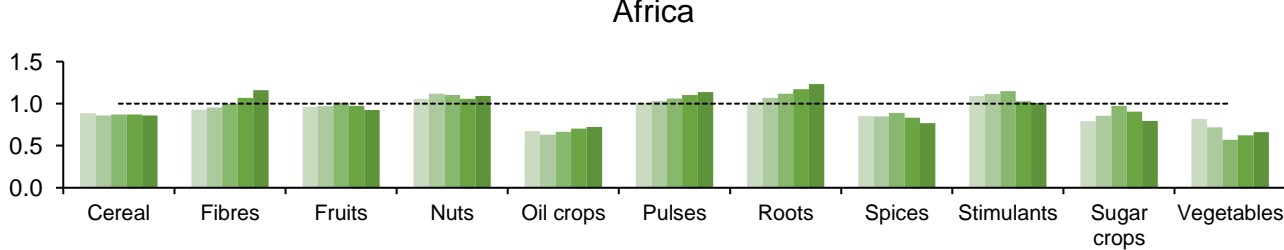

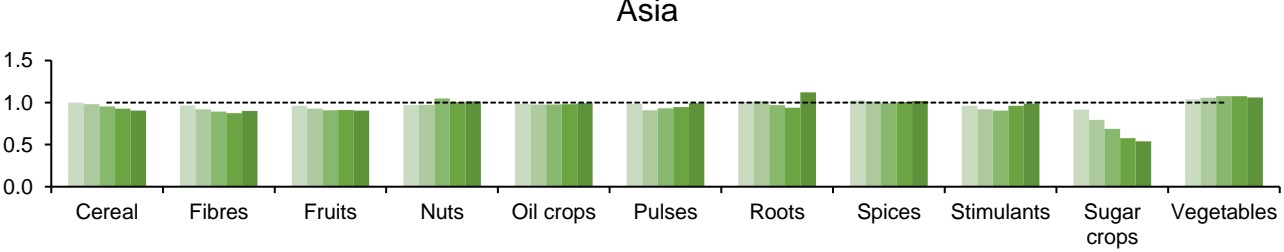

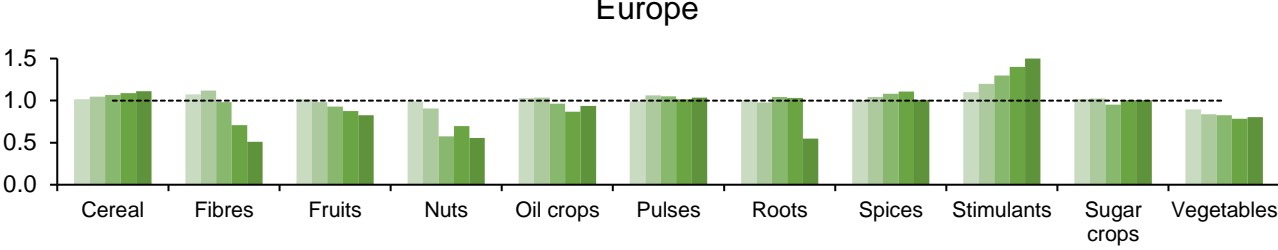

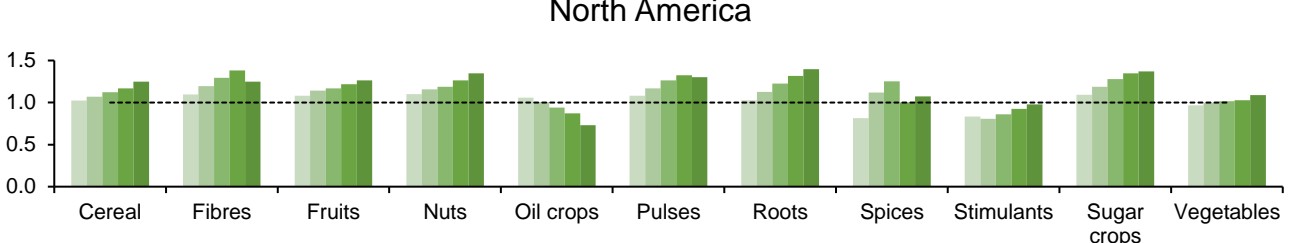

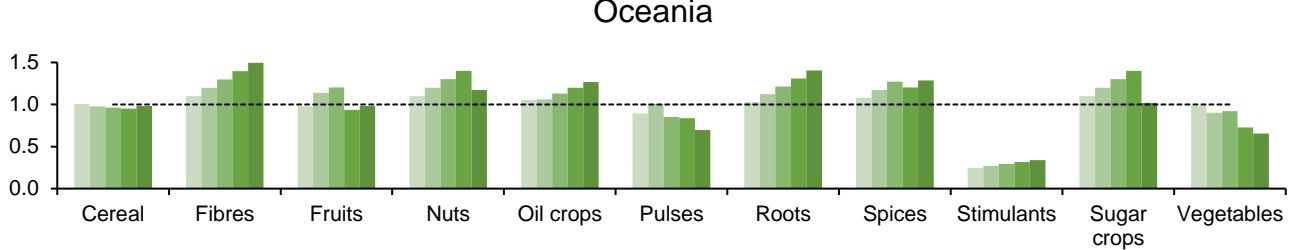

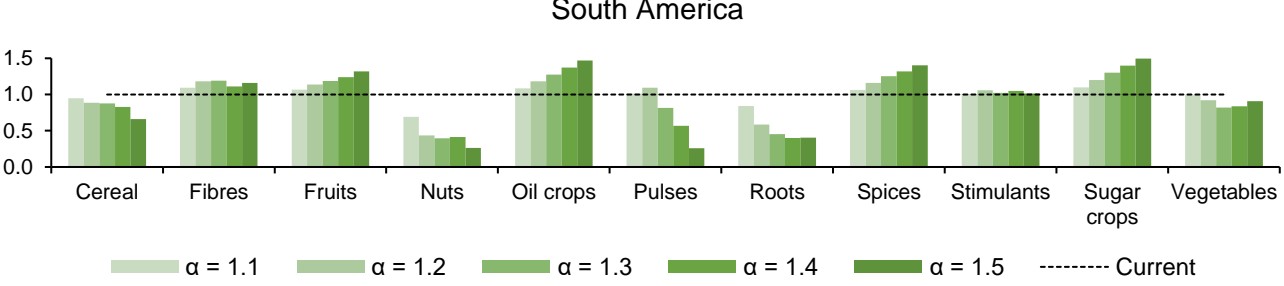

**Figure 3.** Ratio of total production in the optimized cropping pattern to total production in the reference cropping pattern (period 1996-2005), per crop group and per continent, for α = 1.1 to α = 1.5.

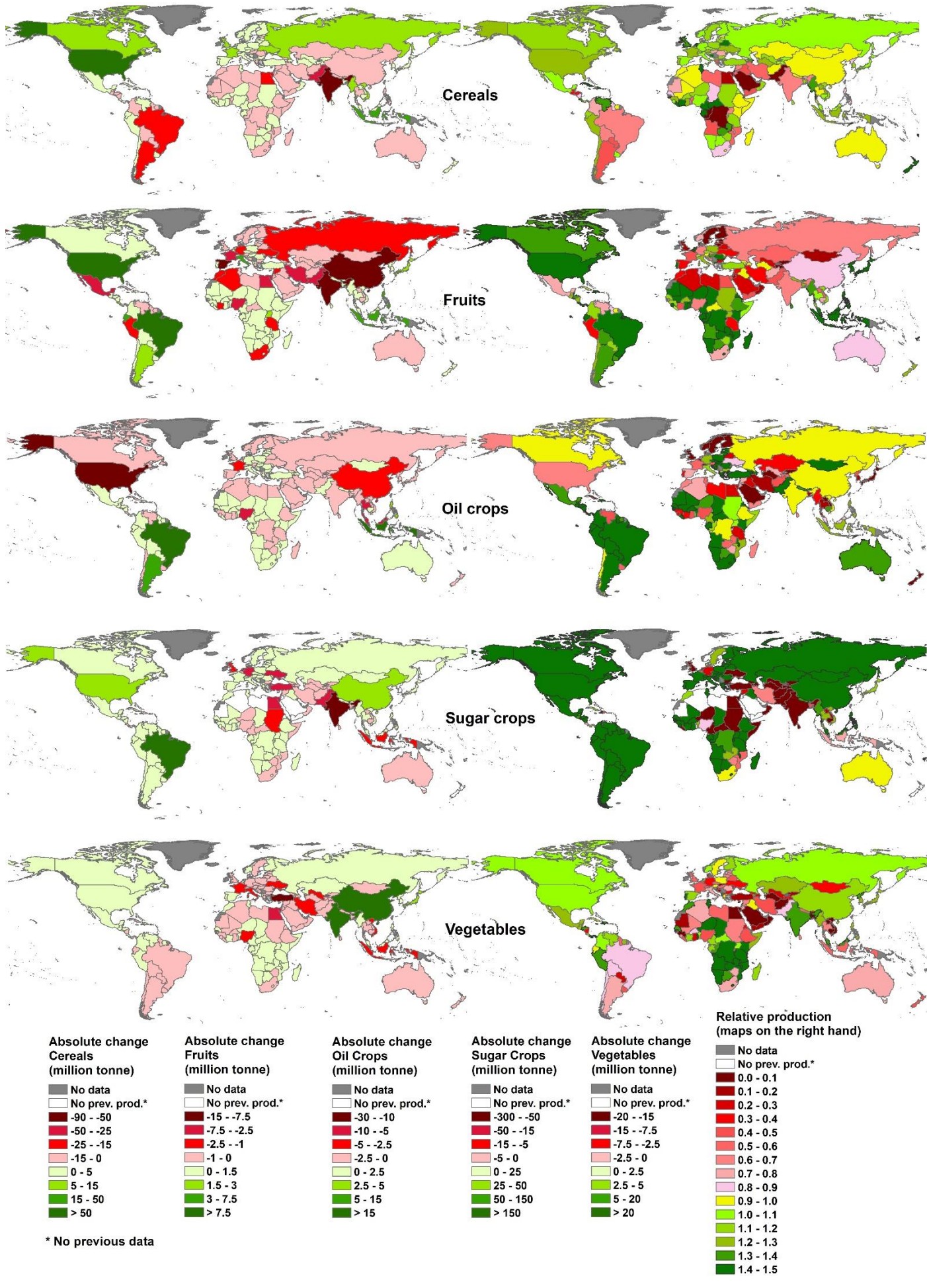

**Absolute change Cereals (million tonne)**

- No data
- No prev. prod.*
- -90 - -50
- -50 - -25
- -25 - -15
- -15 - 0
- 0 - 5
- 5 - 15
- 15 - 50
- > 50

* No previous data

**Absolute change Fruits (million tonne)**

- No data
- No prev. prod.*
- -15 - -7.5
- -7.5 - -2.5
- -2.5 - -1
- -1 - 0
- 0 - 1.5
- 1.5 - 3
- 3 - 7.5
- > 7.5

**Absolute change Oil Crops (million tonne)**

- No data
- No prev. prod.*
- -30 - -10
- -10 - -5
- -5 - -2.5
- -2.5 - 0
- 0 - 2.5
- 2.5 - 5
- 5 - 15
- > 15

**Absolute change Sugar Crops (million tonne)**

- No data
- No prev. prod.*
- -300 - -50
- -50 - -15
- -15 - -5
- -5 - 0
- 0 - 25
- 25 - 50
- 50 - 150
- > 150

**Absolute change Vegetables (million tonne)**

- No data
- No prev. prod.*
- -20 - -15
- -15 - -7.5
- -7.5 - -2.5
- -2.5 - 0
- 0 - 2.5
- 2.5 - 5
- 5 - 20
- > 20

**Relative production (maps on the right hand)**

- No data
- No prev. prod.*
- 0.0 - 0.1
- 0.1 - 0.2
- 0.2 - 0.3
- 0.3 - 0.4
- 0.4 - 0.5
- 0.5 - 0.6
- 0.6 - 0.7
- 0.7 - 0.8
- 0.8 - 0.9
- 0.9 - 1.0
- 1.0 - 1.1
- 1.1 - 1.2
- 1.2 - 1.3
- 1.3 - 1.4
- 1.4 - 1.5


**Figure 4.** Absolute change in production for cereals, fruits, oil crops, sugar crops and vegetables per country (in $10^6$ t/yr) (maps
on the left hand) and relative production (ratio of production in optimized and reference situation) for the same crops groups for
the case of an optimized cropping pattern with $\alpha = 1.5$ (maps on the right hand), all compared to the reference cropping period
(1996-2005): relative production = 1: no change, relative production < 1: countries production is reduced and relative production
> 1: countries production is expanded.

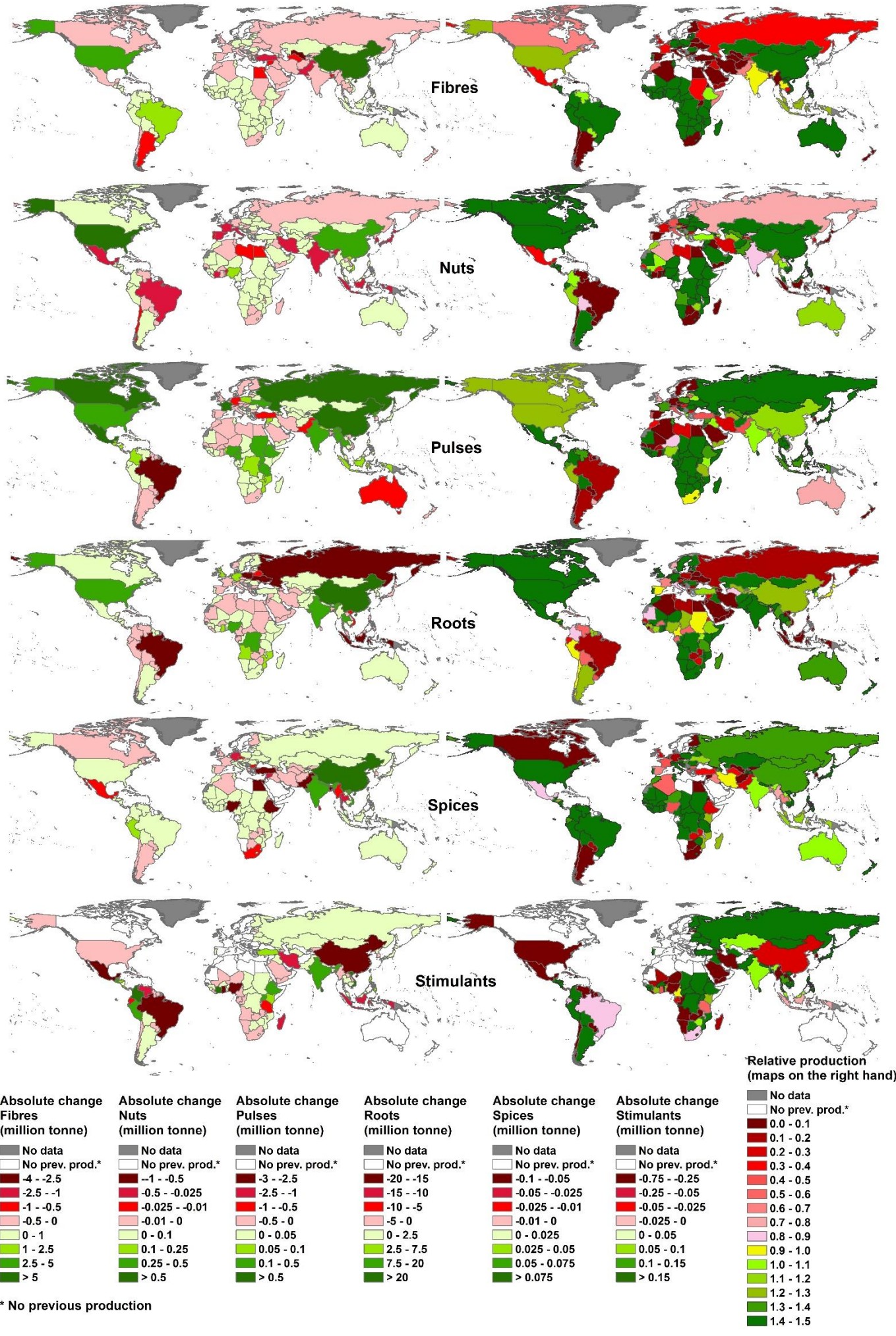

Fibres

Nuts

Pulses

Roots

Spices

Stimulants

| Absolute change Fibres (million tonne) | Absolute change Nuts (million tonne) | Absolute change Pulses (million tonne) | Absolute change Roots (million tonne) | Absolute change Spices (million tonne) | Absolute change Stimulants (million tonne) |
|---|---|---|---|---|---|
| No data | No data | No data | No data | No data | No data |
| No prev. prod.* | No prev. prod.* | No prev. prod.* | No prev. prod.* | No prev. prod.* | No prev. prod.* |
| -4 - -2.5 | --1 - -0.5 | -3 - -2.5 | -20 - -15 | -0.1 - -0.05 | -0.75 - -0.25 |
| -2.5 - -1 | -0.5 - -0.025 | -2.5 - -1 | -15 - -10 | -0.05 - -0.025 | -0.25 - -0.05 |
| -1 - -0.5 | -0.025 - -0.01 | -1 - -0.5 | -10 - -5 | -0.025 - -0.01 | -0.05 - -0.025 |
| -0.5 - 0 | -0.01 - 0 | -0.5 - 0 | -5 - 0 | -0.01 - 0 | -0.025 - 0 |
| 0 - 1 | 0 - 0.1 | 0 - 0.05 | 0 - 2.5 | 0 - 0.025 | 0 - 0.05 |
| 1 - 2.5 | 0.1 - 0.25 | 0.05 - 0.1 | 2.5 - 7.5 | 0.025 - 0.05 | 0.05 - 0.1 |
| 2.5 - 5 | 0.25 - 0.5 | 0.1 - 0.5 | 7.5 - 20 | 0.05 - 0.075 | 0.1 - 0.15 |
| > 5 | > 0.5 | > 0.5 | > 20 | > 0.075 | > 0.15 |

* No previous production

**Relative production (maps on the right hand)**

- No data
- No prev. prod.*
- 0.0 - 0.1
- 0.1 - 0.2
- 0.2 - 0.3
- 0.3 - 0.4
- 0.4 - 0.5
- 0.5 - 0.6
- 0.6 - 0.7
- 0.7 - 0.8
- 0.8 - 0.9
- 0.9 - 1.0
- 1.0 - 1.1
- 1.1 - 1.2
- 1.2 - 1.3
- 1.3 - 1.4
- 1.4 - 1.5


**Figure 5.** Absolute change in production for fibres, nuts, pulses, roots, spices and stimulants per country (in $10^6$ t/yr) (maps on the left hand) and relative production (ratio of production in optimized and reference situation) for the same crops groups for the case of an optimized cropping pattern with α=1.5 (maps on the right hand), all compared to the reference cropping period (1996-2005): relative production = 1: no change, relative production < 1: countries production is reduced and relative production > 1: countries production is expanded.

*Cereal production*. The US and to a lesser extent Indonesia and France have a large absolute and relative changes (Figure 4) for cereals and thus a comparative advantage (given the combination of their water endowments and water productivities compared to other countries). In the case of $\alpha = 1.5$, cereal production of the US, Indonesia and France will increase by 30, 26 and 23%, respectively, compared to the reference situation. India has a comparative disadvantage in cereals and will reduce its production by 40% in the optimized cropping pattern with $\alpha = 1.5$. Looking at the main cereal crops separately (wheat, barley, maize and rice) and combining information on relative and absolute changes, we find that France and the Russian Federation have a comparative advantage in wheat production, with large absolute increases when we optimize the global cropping pattern (Supplemental Figure 1). India and China, contributing 12% and 17% respectively of global wheat production in the reference period, have a comparative disadvantage and shrink their wheat production by 41% for China and 26% for India when $\alpha = 1.5$. For barley, we find Canada, France, Spain, and Turkey to have a comparative advantage. Germany and the Russian Federation, contributing 9% and 11% respectively to the global barley production in the reference period, have a comparative disadvantage and will decrease their barley production respectively by 28% and 88% when $\alpha = 1.5$. For maize, the US is found to have a comparative advantage, while, Brazil, contributing 6% to global maize production in the reference period, has a comparative disadvantage and will reduce its maize production with 64% in the optimized situation ($\alpha = 1.5$). For rice, China, Indonesia and Vietnam have a comparative advantage, with shares in global rice production raising from 32%, 9% and 5% respectively in the reference situation to 22%, 29% and 27% in the optimised situation (when $\alpha = 1.5$). India, contributing 22% to global rice production in the reference period, has a comparative disadvantage and will decrease its rice production with 43% when $\alpha = 1.5$ compared to the reference situation.

*Fruit production.* Comparative advantages for fruit production are found for Brazil and the US, which will increase their respective shares in global fruit production from 7% and 6% in the reference situation to 11% and 9% in the optimized cropping pattern (when $\alpha = 1.5$). China and India, contributing 14% and 10% respectively to global fruit production in the reference period, appear to have a comparative disadvantage and will reduce their fruit production by 13% and 31% respectively in the optimized situation (when $\alpha = 1.5$). Zooming in to the top-4 produced fruits – apples, bananas, grapes and oranges – we find the following. For apples, the US has a comparative advantage; the country will increase its share in global apple production from 8% (reference) to 12% (when $\alpha = 1.5$). China, contributing 35% to the global apple production in the reference period, has a comparative disadvantage and will decrease its apple production by 12% in the optimized cropping patterns (when $\alpha = 1.5$). For bananas, Ecuador, Brazil and the Philippines have a comparative advantage. India, contributing 22% to global banana production in the reference, have a comparative disadvantage. For grapes, Italy, the US and China have a

comparative advantage, with shares in global grape production rising from 15%, 9% and 7% (reference) to 22%, 13% and 10%
($\alpha = 1.5$). France and Spain, contributing 13% and 9% respectively to the global grapes production in the reference situation,
have a comparative disadvantage and will entirely abandon grapes production when $\alpha = 1.5$. For oranges, Brazil and the US
have a comparative advantage, while Mexico, Spain and Iran have a comparative disadvantage (Supplemental Figure 2).
*Oil crops.* For oil crops, we find Indonesia, Brazil and Argentina to have a comparative advantage. Their shares in global
oil crops production will raise from 13, 9% and 6% respectively (reference) to 16, 13% and 9% ($\alpha = 1.5$). The US and
Malaysia contributing 17%, and 12% respectively to global oil crops production in the reference situation, have a comparative
disadvantage and will reduce their oil crops production by 32% and 14% respectively in the optimized cropping pattern (when
$\alpha = 1.5$). Focussing on soybean, which contributes 36% to the global oil crops production, we find the comparative advantage
for Argentina and Brazil. The share of Argentina and Brazil in global soybeans production will rise from 14% and 22%
respectively (reference) to 21 and 33% ($\alpha = 1.5$). China and the US have a comparative disadvantage in soybeans production.
While the US, contributing 43% to the global soybean production in the reference period, will reduce its production by 31%,
China, contributing 9% in the reference period, will entirely stop its soybean production in the optimized pattern (when $\alpha = $
1.5) (Supplemental Figure 3).
*Sugar crops.* Brazil and China have a comparative advantage in sugar crops production, with shares in global sugar crops
production rising from 23% and 6% respectively (reference) to 35% and 9% (optimized cropping pattern with $\alpha = 1.5$). India,
currently contributing 18% to the global sugar crops production, has a comparative disadvantage and will quit sugar crops
production almost entirely. Considering sugar beet and sugar cane separately, we find that France, Poland, the Russian
Federation and the US have a comparative advantage in sugar beet production. Germany, Turkey and Ukraine, contributing
11%, 7% and 6% to the global sugar beet production (reference), have a comparative disadvantage and will decrease their
sugar beet production by 72%, 100% and 94% respectively (when $\alpha = 1.5$). For sugar cane, Brazil and China have a
comparative advantage; their shares in global sugar cane production will increase from 28% and 6% respectively (reference) to
42% and 10% (optimized cropping pattern with $\alpha = 1.5$). India, contributing 22% to global sugar cane production in the
reference period, has a comparative disadvantage and will decrease its sugar cane production by almost 100% (Supplemental
Figure 3).
*Vegetables.* China and India have a comparative advantage in vegetable production. Their shares in global vegetable
production will rise from 45% and 9% respectively (reference) to 52 and 12% respectively (optimized cropping pattern
with $\alpha = 1.5$). Turkey, contributing 4% to global vegetable production in the reference, has a comparative disadvantage and
will reduce its vegetable production by 83% in the optimized pattern (when $\alpha = 1.5$) compared to the reference situation.
Looking at the most produced vegetable crop, tomato, which contributes 15% to global vegetable production, we find that
China and the US have a comparative advantage (Supplemental Figure 3). The share of China and the US in the global

production of tomatoes will increase from 21% and 11% respectively (reference) to 30% and 16% respectively (when $\alpha =$ 1.5). Egypt and Turkey, contributing 6% and 8% to global tomatoes production in the reference, have a comparative disadvantage and will stop their production almost entirely in the optimized situation.

**Sensitivity to restricting expansion to rainfed areas**

By allowing only rainfed areas per crop to expand up to 10%, and irrigated area per crop only to shrink, global blue water consumption of crop production is reduced by 16%. When α is equal to 1.3, 1.5 and 2.0 (i.e. when harvested area per crop per country can expand by up to 30%, 50% and 100%), global blue water consumption gets reduced by 31%, 41% and 54%, respectively. The maximum blue water scarcity is reduced to a scarcity of 62%, 14%, 5% and 3% when α equal to 1.1, 1.3, 1.5 and 2.0 respectively (Table 4).

**Table 4.** Current versus optimized maximum BWS when allowing both irrigated and rainfed areas to expand and when allowing only rainfed areas to expand and the share of rainfed areas sifts in reducing maximum BWS for the case when $\alpha$ equal to 1.1, 1.3, 1.5 and 2.0 respectively

| | | Maximum BWS | | Reduction in maximum BWS compared to reference situation | | Share of rainfed shifts in reducing maximum BWS |
| | | Optimized | | | | |
| Factor α | Current* | Expansion in both irrigated and rainfed areas | Expansion in only rainfed areas | Expansion in both irrigated and rainfed areas | Expansion in only rainfed areas | |
| --- | --- | --- | --- | --- | --- | --- |
| $\alpha = 1.1$ | 272% | 39% | 62% | -86% | -77% | 90% |
| $\alpha = 1.3$ | 272% | 6% | 14% | -98% | -95% | 97% |
| $\alpha = 1.5$ | 272% | 4% | 5% | -99% | -98% | 99% |
| $\alpha = 2.0$ | 272% | 2% | 3% | -99% | -99% | 99.6% |

\* independent of $\alpha$

The shifts in only the rainfed area give a dominant contribution to the reduction of the maximum BWS in the case when allowing both rainfed and irrigated areas to expand. Contributions from only rainfed shifts amount to 90% of the total reduction when α equal to 1.1 to 97, 99 and 99.6% when α equal to 1.3, 1.5 and 2.0 respectively. The dominance effect of shifts in rainfed areas proves that the optimization results are not very sensitive to modest allowed expansion in irrigated areas per crop.

**Discussion**


Our study has some limitations that need careful consideration in interpreting results. Limited by availability of some of
the required data and operational computational limitations of optimization software, we analyse the global cropping pattern at
the country scale rather than at sub-national or grid-scale. However, having a high average yield for a specific crop in a certain
country doesn't necessarily mean that everywhere in that country the same performance in terms of land and water
productivity is achieved, due to spatial differences in crop suitability. This could result in reallocating crops to countries that
have a very limited suitable production area but are productive in terms of water and land in the reference situation. To
constrain this effect, we do not allow total cropland per country to expand, so that areal expansion for one crop replaces the
land use of another crops with a shrinking area; also, we limit the expansion in cropland by a certain maximum rate for each
crop per country (the factor $\alpha$). The analysis at country level also has implications for the interpretability of water scarcity
indicators. Assessing water scarcity at the level of a country level and an average year hides the water scarcity that manifests
itself in particular places within countries or on particular periods (Mekonnen and Hoekstra, 2016). We minimize *average*
water scarcity in countries; within countries scarcity differences will still appear, both in the reference situation and in the case
of the optimized cropping patterns. Still, water scarcity indicators at national levels provide insight; within the framework of
the Sustainable Development Goals, indicator 6.4.2 (Level of water stress), is used to monitor Goal 6 (Ensure availability and
sustainable management of water and sanitation for all); it is defined similar to water scarcity in our study, also at the
resolution of countries, but based on water extractions rather than consumptive water use. Where lowering the water stress
level is a goal for each country, from a global equity perspective lowering stress in countries with highest water scarcity is
prioritised. This is operationalised by choosing the maximum national water scarcity as an objective function in the
optimization. Relieving water scarcity in specific hotspots within countries by changing cropping patterns could be studied
using the current approach but is beyond the scope of this paper.  The sensitivity analysis did show that by far the largest
impact on water scarcity relief emerges from shifts in cropping patterns of rainfed crops, not depending on the heterogeneity of
blue water availability; therefore water scarcity reduction in countries with highest scarcity at national level in the current
study does not rely on worsening water scarcity in countries with heterogeneous conditions.
Another limitation of this study is the focus on water and land endowments and productivities and on global water
scarcity reduction as a shared goal, while other production factors such as labour, knowledge, technology and capital can be
limiting factors to expand production of certain crops in some countries and certainly agroeconomic aspects may play a role in
considering comparative advantages as well. Other factors could be included in a future study by refining the optimization
model; other objective functions could emphasize trade-offs between economic and environmental goals. Moreover,
agricultural, trade and food security policies could be other factors that drive cropping patterns rather than water and land
availability (Davis et al., 2018). Here, we purposely limited our analysis to considering comparative advantages from a

perspective of land and water resource use to understand the specific role of these two particular factors. By no means we suggest that the 'optimized cropping patterns' found here are 'better' than the reference pattern because what is best depends on a lot more factors than included here, including political preferences. Rather, our results are instrumental in illustrating directions of change if we would put emphasis on the factors land and water endowment and productivity and put particular value to reducing water scarcity in the most water-scarce places.

The scope of the current study is restricted to the exploration of alternative cropping patterns to reduce water scarcity in the reference situation; we therefore use reference resource efficiencies. We do not take into consideration the future increase in food demand due to population growth, nor of agronomic developments that may increase resource use efficiencies, nor of climate change that will affect the future ability of countries to produce crops. The current study supports the findings of Davis et al., (2017a) on the benefits of crop redistribution on water saving which could be a potential strategy for sustainable crop production and an alternative to the large investments that are usually needed to close up the technological and yield gaps in developing nations. Besides reducing water and land use, changing cropping pattern will also have an impact on reducing GHG emission that results from extensive energy activities in irrigation such as groundwater pumping which accounted for around 61% of total irrigation emissions in China (Zou et al., 2015).

The results suggest that Asia, for example, could contribute to global water scarcity mitigation by reducing its production of fruits and sugar crops while increasing its cereal and vegetable production. This implies that Asia will move to economically less attractive crops. This illustrates the possible trade-off between the goal of reducing water scarcity in the most water-scarce countries and the goal of economic profit by producing cash crops by individual countries or regions. The optimization results do not pretend that the changes in production patterns are likely to occur, but merely that these changes reduce water scarcity most; national and international policies would be required to promote such water-saving changes to be implemented (Klasen et al., 2016).

Changing cropping patterns could reduce global blue water footprint by 21% and global irrigated area by 10%. These findings prove that current high scarcity levels in a serious number of countries is shown to be caused by the current crop allocation pattern, rather than by an inevitability of those scarcities to occur; that suggests that water endowment is insufficiently driving crop allocation to avoid water scarcity. This in consistent with Zhao et al., (2019) who find in their study for China that comparative advantages with respect to labour and water were not reflected in the regional distribution of agricultural production. However, not all countries would benefit similarly in the optimized set, India and China, main crop producers in the reference situation, will only start to have a decrease in their blue water scarcity when the allowed expansion rate is larger than 20%. This is in line with the findings of Davis et al., (2017a) who find in their simulations that water scarcity persists in many important agricultural areas (the US Midwest, northern India, Australia's Murray-Darling Basin, for example), indicating that extensive crop production in these places prohibits water sustainability, regardless of crop choice (Davis et al., 2017a).

Findings suggest that China, one of the main producers of the major crop in the world, will abandon soybean production and halve wheat irrigation area. This will relieve some of the pressure on the northern part of China where water scarcity is the most severe (Ma et al., 2020). China will increase the harvested area of rice and rapeseed, the crops with the most significant comparative advantage in terms of land and water. Similarly, our results suggest that the US, another major crops producer, would and restrict soybean production to rainfed systems, abandoning irrigation, in the optimized set in the US. The US focuses on producing maize, mainly rainfed, for which the US has a comparative advantage in terms of water and land productivities. This may be a great relief to the US corn belt where most of irrigated soybeans and maize are located (Zhong et al., 2016) and could be a remedy to the projected water shortage of that region resulting from population growth and climate change (Brown et al., 2019). We also find that India, another major producer of crops in the world, will move away from sorghum production and shift a large share of its rice and wheat production to rainfed conditions. Moving to rainfed production in India could mitigate the effect of the intensive use of irrigation from groundwater and surface water which caused groundwater degradation in many districts of Haryana and Punjab, the largest contributing states to rice and wheat production in India (Singh, 2000).

For some of the most water-scarce countries, results show that blue water consumption in crop production is reduced by more than 70% compared to the reference situation: Cyprus, Egypt, Iran, Jordan, Kuwait, Libya, Pakistan, Saudi Arabia, Syria, Turkmenistan and Yemen. This means that these countries, with modest rainfed agricultural areas, will rely more heavily on imports and thus become highly dependent on other countries. Most of these countries already have a high dependency on crop imports in the reference situation. This reflects a trade-off between reducing water scarcity and increasing food security on the one hand and shows the important role of food trade in relieving water scarcity in many places in the world on the other.

**Conclusion**

When allowing a 10% maximum expansion of harvested area per crop and per country, while not allowing an increase in total rainfed or irrigated cropland per country, a global blue water saving in the world of 170,000 million m$^3$/yr is achievable, which is 21% of the current global blue water footprint. Changes in the cropping pattern of rainfed production have a dominant effect, relieving irrigated areas to contribute to production; the total global harvested area would decrease by 2% while the total global irrigated area would decrease by 10%. The blue water scarcity in the seven countries with highest national water-scarce, Libya, Saudi Arabia, Kuwait, Yemen, Qatar, Egypt, and Israel (with current scarcities ranging from 54% to 270%), can be reduced to a scarcity of 39% or less. Optimizing the global cropping pattern to reduce the highest national water scarcity comes with trade-offs, where severely water-scarce countries reduce water scarcity at the expense of decreased food self-sufficiency.

When considering how to change the global cropping pattern in order to reduce water scarcity in the world's most severely water-scarce countries, we specifically find the following major shifts. Cereal production is reduced in Africa and South America and increased in North America and Europe. Fruits production is reduced most significantly in Asia and Africa and expanded in the Americas. Oil crops production is reduced most significantly in Africa and expanded in the Americas. Sugar crop production is reduced most significantly in Asia and Africa and expanded in the Americas. Vegetable production is reduced most significantly in Europe and Africa and expanded in Asia. Reallocating cereal crops is the main contributor to global blue water saving with a contribution of 50% for the case of α = 1.1, followed by fruit, sugar crops and fibres with 12%, 10% and 9% respectively.

From a water and land perspective and aiming for global water scarcity reduction, comparative advantages for cereal production are found for the US and to a lesser extent Indonesia and France, whereas India has a comparative disadvantage. The comparative advantage of the US refers to maize, for France to Wheat and Barley and for Indonesia to rice. India's comparative disadvantage in cereal production particularly refers to wheat and rice. For fruit production, Brazil and the US are found to have a comparative advantage, whereas China and India have a comparative disadvantage. More in particular, the US has a comparative advantage for apples, grapes and oranges, and Ecuador and Brazil for banana, while China has a comparative disadvantage in apples, and India for bananas. For oil crops, Indonesia, Brazil and Argentina have a comparative advantage, and the US and Malaysia a comparative disadvantage. Argentina and Brazil have a comparative advantage for soybean, while the US and China have a comparative disadvantage. For sugar crops production, Brazil and China are found to have a comparative advantage, while India have comparative disadvantage for sugar crops. Brazil and China have a comparative advantage for sugar cane, while India has a comparative disadvantage for sugar cane. For vegetables, we find China and India to have a comparative advantage and Turkey to have a comparative disadvantage. China has a comparative advantage for tomatoes and Turkey a comparative disadvantage.

By considering differences in national water and land endowments, following the Heckscher-Ohlin (H-O) theory of comparative advantage, as well as differences in national water and land productivities, following Ricardo's theory of

comparative advantage, we combine two rationales that are both relevant. With the optimization exercises carried out in this
study we show that blue water scarcity can be reduced to reasonable levels throughout the world by changing the global
cropping pattern, while maintaining current levels of global production and reducing land use.
**Data availability**
The datasets generated during and/or analysed during the current study are available in the supplementary information and the
4TU.ResearchData repository (CC-BY-NC-ND), https://doi.org/10.4121/uuid:64e7f59a-03f3-4e25-83c8-06745e9216d2.
**Author contribution**
The three authors designed the research, analysed the data and wrote the paper. H.C carried out the calculations.
**Competing interests**
The authors declare that they have no conflict of interest.

**Acknowledgements**
H.C. and M.S.K. dedicate this work to their co-author prof. dr. Arjen Hoekstra who passed away unexpectedly just before the
revision of this paper and whose ideas and ideals greatly influenced and will still influence a generation of scientists. The work
by M.S.K. and A.Y.H was partially funded by the European Research Council (ERC) under the European Union's Horizon
2020 research and innovation programmes, projects "Moving Towards Adaptive Governance in Complexity: Informing Nexus
Security" (MAGIC), Grant Agreement No. 689669 and Earth@lternatives, Grant Agreement No 834716.

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
