# Peer review of "Changing global cropping patterns to minimize national blue water scarcity"

_Hydrology and Earth System Sciences, 2019_

## Referee Comment (RC1) · Anonymous Referee #1 · 12 Nov 2019

General comments

The authors determine for a large number of crops how crop production could be shifted among the countries of the world to produce the same amount of each crop globally while minimizing the highest value of a country-scale indicator of blue water scarcity, without any extension of the total national cropland but a with a certain maximum allowed extension of cropping area in the countries, both for rainfed and irrigated production.

Mainly for reasons described as limitations of the study by the authors themselves (lines 368-378 but also 379-383) I think that the results of the study are not informative

and even misleading. This is due to the scale of the study which inclusively considers countries as homogeneous units of analysis, regarding land and water productivities as well as blue water availability.

The novelty claimed in the manuscript is consideration of blue water scarcity. Unfortunately, blue water scarcity is only considered as one value per country, computed as the ratio of total blue water use in the the country and blue water availability in the country. This is problematic as their are important crop-producing large countries like India, China and the US (but also Australia) with humid and semi-arid climate zone, where irrigated crop production and thus blue water use is concentrated in the semi-arid/arid regions of the country while blue water availability is high the humid parts of the country. This is why these countries, in which large regions suffer from irrigation-induced water stress and even groundwater depletion, do not appear among the 21 countries with the highest water scarcity (Table 2) for which the authors show to what extent blue water consumption and thus blues water scarcity could be reduced by shifting crop production to other countries (with lower blue water scarcity). One result is that in the optimized distribution of crop production among countries, both China, India and Australia increase their blue water consumption (Fig. 2 bottom). I do not find it plausible that the thus optimized distribution of crop production among countries "minimizes blue water scarcity in the worlds's hotspots" (as is formulated in the title).

I think it is a prerequisite for publication of the study that the authors show the results of a sensitivity analysis regarding the spatial analysis units. Blue water availability values as well as irrigated areas are available at a spatial resolution of $0.5°$ by $0.5°$, and this information could be used to see how the optimization results change if the blue water availability in the irrigated areas/cropping areas are taken into account instead of average country values. You could have a look at Yano et al. 2015 (Yano S, Hanasaki N, Itsubo N and Oki T 2015 Water scarcity footprints by considering the differences in water sources Sustainability 7 9753–72) where water scarcity at the country and for irrigated areas are computed separately and compared. Blue water availability from

various global hydrological model available at www.isimip.com could be used.

In addition, it is necessary to broaden the literature review. For example, the work of Taikan Oki and his group have not been considered. Please review Oki et al. 2017, Environ. Res. Lett. 12 044002 and some of the references therein. Oki and Kanae 2004 already showed global water savings by global trade.

Specific comments

L76: Jalava et al. 2016 also studied the effect of food loss reduction (https://doi.org/10.1002/2015EF000327)

L79: Explain more clearly to a broader audience what the definition of virtual water is (also: does not only relate to food).

L102ff. Explain more clearly the study of Davis et al. 2017a, and compare their methods and results to your study (e.g. in the discussion section). L111: Define clearly here that "cropping patterns" mean the distribution of production of a certain crop among the nations/countries but not within.

L118: Expand methods section with respect to considered crops/crop groups, algorithm for optimization, e.g. how was ensemble of potential cropping patterns produced?

L139: BWS only takes into account irrigation water use but not the other use sectors. Define blue water footpring.

L159: Explain why you chose to minimize (only) the highest national blue water scarcity.

L220-364. Please shorten the lengthy description of the changing cropping patterns and comparative advantages shown in figures and tables but try to explain the results.

L367ff Also discuss the real-life meaning and consequences of optimized global cropping pattern, in particular reduced blue water consumption in the countries listed in Table 2. E.g. if BWC is reduced from 1900 to 280 million m3/yr in Libya, crop production (Fig. 4) and income would be strongly reduced, too. Could the production/income

loss be somehow related to GDP to understand the problems that would result from the analyzed global-scale optimization?

L408 ff. I would not use the grammatic form of "will", e.g. in "Cereal production will get reduced in Africa". Maybe better: "If blue water scarcity was globally optimized, cereal production would be reduced in Africa according to our analysis."

---

## Referee Comment (RC2) · Anonymous Referee #2 · 6 Dec 2019

The study on "Changing global cropping patterns to minimize blue water scarcity in the world's hotspots" provides a new view of possibility to reduce crop-related blue water footprint and diminish the severe blue water scarcity worldwide. Plenty work has been done in this study, however, I feel that some parts in the text require careful revisions and improvements before it can be further considered for publication in HESS.

1. Line 31, you mentioned in the abstract 'changing spatial cropping patterns and international crop trade...", but just showing the 'spatial cropping patterns' changes. It could be much better to look at further on hotspot countries in terms of the responses in trade patterns (just changes in crop trade balances).

[Figure]

2. Line 111, in the introduction of study content, information on how many types of crops considered is lacking.

3. Line 112-113, the first and second constrains seems conflict each other.

4. How do you define the 'cropping pattern' ïïj§

5. In the analysis, how the green water limits were considered? I am wondering if there are some places with increasing green WFs but have insufficient green water availability?

6. Line 213, for China you show an 4% increase in BWC. It looks tiny for the whole country, but could matter when such increases in BWC happend in a very severe blue water scarce places within the country. At least some discussion regarding this should be in somewhere of the text. In addition, I am also worry about the assumptions of increasing harvested area per crop so that it could resulted in increases in harvested area in each country, or I could be wrong in understanding the first assumption. Given that for example in China, the national policy is controlling not reducing the total crop harvested area to a level with no possibility to increase anymore... The issue is also important for developing countries facing rapid urbanization in land. Maybe better to discuss this in some points.

7. I get confusions when reading the Discussion. It looks too much limitations to get published, too 'optimized' beyond the real. It may be nice to look into the mass of results and pick some countries with results that really meaningful for local national water management. Please carefully consider about how to interprate in the discussion part. Another limitation should be in caution is the issue related to green water availability, scarcity and limits.

minor comments: 1. Line 61, better to give the full name of WEF, either in the reference list. 2. Table 1, the initial sources of harvested areas or productions should be listed as well.

---

## Author Comment (AC1) · 13 Dec 2019

Note to the editor and reviewers:

With deep sadness, we would like to inform you that Arjen Y. Hoekstra, one of the authors of this paper, has passed away on the 18th of November 2019. We will maintain Arjen as co-author for this paper for all his valuable contributions in the writing phase of the paper until submission.

Hatem Chouchane and Maarten Krol.

[Figure]

461, 2019.

---

## Author Comment (AC2) · 13 Dec 2019

**Response to comments from reviewer RC1**

Note that reviewer's comments are in italic black, and responses in plain blue font.

General comments:

*The authors determine for a large number of crops how crop production could be shifted among the countries of the world to produce the same amount of each crop globally while minimizing the highest value of a country-scale indicator of blue water scarcity, without any extension of the total national cropland but a with a certain maximum allowed extension of cropping area in the countries, both for rainfed and irrigated production. Mainly for reasons described as limitations of the study by the authors themselves (lines 368-378 but also 379-383) I think that the results of the study are not informative and even misleading. This is due to the scale of the study which inclusively considers countries as homogeneous units of analysis, regarding land and water productivities as well as blue water availability.*

We thank the reviewer for his critical comments. As the reviewer already noted, most limitations observed in the comments are acknowledged and described in the paper's discussion. The main issue here thus is the extent to which the usage of country-average data and the interpretation of results is appropriate. Firstly, one relevant methodological aspect appears misinterpreted: the allowed land-use changes at country level (limited by factor-alpha) is not an allowable expansion in rainfed / irrigated crop area per country limited by national agricultural area, but rather is an allowable shift in the cropping pattern within the bounds of current rainfed and irrigated area per country. So current production characteristics on currently irrigated lands are not assumed to be valid elsewhere. The modest allowed changes in cropping areas of individual crops prevent significant shifts in crop allocation within a country (e.g. to other agro-ecological zones), avoiding implausible results due to the heterogeneity in rainfed and irrigated land productivity. The impact of the observed heterogeneity in blue water availability can be more influential. Water availability is a complex variable because the same volume of water at a specific location and time can be considered available for use at any downstream location and (if storages are present) at any moment in the year; countries base their water management on these properties and implement policies of water allocation within a river basin, reservoir construction and management and large scale inter-basin water transfers. The extent of such policies to justify only considering total national freshwater availability in assessing water scarcity is limited, however, calling for care in interpreting national-scale water scarcity as an indicator and in performing scenario exercises as in the present manuscript. We agree that this discussion is underemphasized in the paper and will better address it. This discussion closely links to considerations on the choice of *Water stress* (freshwater withdrawal as a proportion of available freshwater resources) at country and region level as

indicator 6.4.2 in the SDG framework (UN-Water, 2018). Next, we will add a variation of the current optimization exercise, contributing to assessing the sensitivity of results to the assumed availability of total renewable freshwater at irrigation areas (see response to the next comment).

It should be noted here, that by far the largest impact on water scarcity relief emerges from shifts in cropping patterns of rainfed crops, not depending on the heterogeneity of blue water availability. The dominance of this aspect of the changed global cropping pattern is illustrated using an additional optimization exercise to separate out this effect (see response to the next comment).

Where the scale of analysis chosen in the paper calls for careful introduction of the definition of the exercise and its interpretation, to our knowledge it considers, for the first time, both differences in water productivities and in water endowments to analyse comparative advantages of countries for different types of crop production.

*The novelty claimed in the manuscript is consideration of blue water scarcity. Unfortunately, blue water scarcity is only considered as one value per country, computed as the ratio of total blue water use in the the country and blue water availability in the country. This is problematic as their are important crop-producing large countries like India, China and the US (but also Australia) with humid and semi-arid climate zone, where irrigated crop production and thus blue water use is concentrated in the semi-arid/arid regions of the country while blue water availability is high the humid parts of the country. This is why these countries, in which large regions suffer from irrigation-induced water stress and even groundwater depletion, do not appear among the 21 countries with the highest water scarcity (Table 2) for which the authors show to what extent blue water consumption and thus blues water scarcity could be reduced by shifting crop production to other countries (with lower blue water scarcity). One result is that in the optimized distribution of crop production among countries, both China, India and Australia increase their blue water consumption (Fig. 2 bottom). I do not find it plausible that the thus optimized distribution of crop production among countries "minimizes blue water scarcity in the worlds's hotspots"(as is formulated in the title).*

*I think it is a prerequisite for publication of the study that the authors show the results of a sensitivity analysis regarding the spatial analysis units. Blue water availability values as well as irrigated areas are available at a spatial resolution of 0.5◦ by 0.5◦ , and this information could be used to see how the optimization results change if the blue water availability in the irrigated areas/cropping areas are taken into account instead of average country values. You could have a look at Yano et al. 2015 (Yano S, Hanasaki N, Itsubo N and Oki T 2015 Water scarcity footprints by considering the differences in water sources Sustainability 7 9753–72) where water scarcity at the country and for irrigated areas are computed separately and compared. Blue water availability from various global hydrological model available at www.isimip.com could be used.*

The above comment closely connects to the first one. We fully agree that the term world's hotspots of water scarcity is formulated too loosely, and will improve on that. We also agree that modest to low water scarcity indicators at national level may hide hotspots within a country; we do note however that still water stress or water scarcity are widely used as indicators for the human pressure on water resources as national scale, e.g. SDG 6.4.2 Water Stress in FAO's Aquastat, intended for country comparisons in global studies.

Results on insight-raising variations to the optimization exercise will be added as sensitivity study; this concern:

- an exercise to identify the impact of shifts in the rainfed cropping pattern only, only allowing irrigated production areas to decrease for each country and crop;
- an exercise to separate out the impact of allowed water scarcity increases in countries with low scarcity, by disallowing increases in blue water use in each country (contrary to the optimization in the current manuscript).

*In addition, it is necessary to broaden the literature review. For example, the work of Taikan Oki and his group have not been considered. Please review Oki et al. 2017, Environ. Res. Lett. 12 044002 and some of the references therein. Oki and Kanae 2004 already showed global water savings by global trade.*

We will broaden the literature review and consider for example the work of Taikan Oki and his group.

*Specific comments*

*L76: Jalava et al. 2016 also studied the effect of food loss reduction (https://doi.org/10.1002/2015EF000327)*

Results will be discussed in the context of this reference; a citation will be added.

*L79: Explain more clearly to a broader audience what the definition of virtual water is (also: does not only relate to food).*

The definition of virtual water has been changed into the following: The trade in 'embedded water' ( also known as virtual water trade) is the hidden flow of water if food or other commodities are traded from one place to another (Allan, 1998).

*L102ff. Explain more clearly the study of Davis et al. 2017a, and compare their methods and results to your study (e.g. in the discussion section).*

We add the following:

In the introduction: "However, the current study has a number of differences with Davis et al. (20017a). First, we consider a larger number of crops (125 crops including vegetables, fruits and pulses which were not considered in Davis et.al., (2017a) study). Second, we are changing cropping patterns while maintaining same global production per crop while Davis et al. (2017b) aim for higher calories and protein production while reducing water use which could result in producing different crops in the optimized set than in the current set."

In the discussion we add:

"The current study supports the findings of Davis et al (2017a) on the benefits of crop redistribution on water saving which could be a potential strategy for sustainable crop production and an alternative to the large investments that are usually needed to close up the technological and yield gaps in developing nations."

"Changing cropping patterns could reduce global blue water footprint by 9%. However, not all countries would benefit similarly in the optimized set, India and China, for example, would have a slight increase in their blue water consumption by 5% and 4% respectively. This is in line with the findings of Davis et al. (2017a) who find in their simulations that water scarcity persists in many important agricultural areas (the US Midwest, northern India, Australia's Murray-Darling Basin, for example), indicating that extensive crop production in these places prohibits water sustainability, regardless of crop choice (Davis et al.2017a)."

*L111: Define clearly here that "cropping patterns "mean the distribution of production of a certain crop among the nations/countries but not within.*

We add the following explanation: "(By changing cropping patterns, we mean that the allocation of crops to rainfed and irrigated land within a country changes, where both rainfed and irrigated area of a certain crop can be expanded up to a modest maximum rate, while respecting the bounds of current total rainfed and total irrigated area per country)".

*L118: Expand methods section with respect to considered crops/crop groups, algorithm for optimization, e.g. how was ensemble of potential cropping patterns produced?*

We add: "We considered 125 crops of the main crops groups (cereals, fibres, fruits, nuts, oil crops, pulses, roots, spices, stimulants, sugar crops and vegetables) (for an extensive list of crops used see (Chouchane et al., 2019)); optimization was performed using the linear optimization routine from the Optimization Toolbox of MATLAB".

*L139: BWS only takes into account irrigation water use but not the other use sectors. Define blue water footpring.*

We added the following: "Blue water footprint (BWF) refers to the volume of consumptive freshwater use for irrigation that comes from surface and groundwater".

*L159: Explain why you chose to minimize (only) the highest national blue water scarcity.*

Within the framework of the Sustainable Development Goals, SDG 6.4.2 (*Level of water stress*), is used as an indicator for *Goal 6. Ensure availability and sustainable management of water and sanitation for all*; it is defined similar to water scarcity here, also at the resolution of countries, but based on water extractions rather than consumptive water use. Where lowering the water stress level is a goal for each country, from a global equity perspective lowering stress in countries with highest water scarcity is prioritised. This is operationalised by choosing the maximum national water scarcity as an objective function in the optimization.

*L220-364. Please shorten the lengthy description of the changing cropping patterns and comparative advantages shown in figures and tables but try to explain the results.*

This will be shortened in the revised version.

*L367ff Also discuss the real-life meaning and consequences of optimized global cropping patteren, in particular reduced blue water consumption in the countries listed in Table 2. E.g. if BWC is reduced from 1900 to 280 million m3/yr in Libya, crop production (Fig. 4) and income would be strongly reduced, too. Could the production/income loss be somehow related to GDP to understand the problems that would result from the analyzed global-scale optimization?*

Consequences of the changes in the global cropping pattern on agricultural economy, farm economy and food self-sufficiency are outside of the scope of this paper. Changes towards the optimized cropping patterns identified here would require agroeconomic policies, e.g. on commodity prices, price- and farm income subsidies or trade regulations to reflect implicit resource use.

We already mentioned some impacts related to reduced production in real life. We mentioned the countries with the largest decrease in their blue water footprint of crop production (last paragraph in the

discussion) and the impact that could result from that. However, this doesn't mean directly that the total production is reduced. Since for some countries, when possible, they will switch to rainfed production. So, income reduction is not necessarily proportional to the reduction in blue water consumption. To be able to assess the impact of the reduction in BWC on the country GDP we should be able to trace back the consumption per crop per country and initial import and export. By calculating the changes in consumption, import and export we could assess the changes in the GDP. This is out of the paper scope for now.

*L408 ff. I would not use the grammatic form of "will", e.g. in "Cereal production will get reduced in Africa". Maybe better: "If blue water scarcity was globally optimized, cereal production would be reduced in Africa according to our analysis."*

We will improve the grammatic form through the paper.

References

Chouchane, H., Krol, M. S., and Hoekstra, A. Y.: Dataset for Changing global cropping patterns to minimize blue water scarcity in the world's hotspots, TU.Centre for Research Data, Dataset, https://doi.org/10.1016/j.wroa.2018.09.001, 2019.

UN-Water: Progress on Level of Water Stress – Global baseline for SDG indicator 6.4.2. UN-Water, Geneva, Switzerland, 2018.

---

## Referee Comment (RC3) · Anonymous Referee #3 · 15 Dec 2019

The research on "Changing global cropping patterns to minimize blue water scarcity in the world's hotspots" used a linear optimization algorithm to assess how to change global cropping patterns to reduce blue water-scarce hotspots, with the constraints of global production per crop and current cropland areas. Below are my comments and suggestions:

1. The linear optimization algorithm is set for an optimal reduction of blue water scarcity by changing global spatial cropping patterns. The algorithm set an upper limit of the expansion in cropland by a certain maximum rate for each crop per country (the factor ð İŽij), and also limit total cropland to the reference extent. However, there is no lower

limit of decrease in cropland area, which means cropland area (or crop production) for some crop types would decrease a lot or even disappear (as shown in results part). Why you set an upper limit, but without a lower limit? If you also set both upper and lower limits of changes in cropland for each crop, do the results change?

2. Blue water scarcity (BWS): BWS is defined as the total blue water footprint divided by the blue water availability in the country. Here blue water footprint only includes agriculture sector, without water footprint for domestic and industrial. Blue water availability is the natural runoff, which follows Hoekstra et al. (2012), right?

3. L145: "A country is considered to be under low, moderate, significant or severe water scarcity when BWS is lower than 20%, in the range 20-30%, in the range 30-40% and larger than 40%, respectively (Hoekstra et al., 2012)". Hoekstra et al (2012) analyzed the BWS at basin level and monthly time scale. But this study assesses water scarcity at country level and annual time scale, I think more discussion is needed to illuminate whether the index used here is suitable.

4. L148: why you choose maximum national blue water scarcity in the world as the indicator for optimization?

5. There are too much results about the changing cropping patterns and comparative advantages. I think the authors could add more explanation on the mechanism behind the changes, especially for some typical countries.

6. Discussion part: Previous studies have done a lot of works on the impacts of changing cropping patterns, international food trade and better water productivity on water scarcity (as list in introduction part). I think the discussion part should add more about the similarity and difference between the results in this study and previous studies.

7. More discussions should focus on how the results represented in this study could guide global international food trade, as well as cropping patterns to cope with global water scarcity, especially under future climate change and socioeconomic development. For example, blue water scarcity would intensify in the future as reported in previous studies. And following the results in this study, a water-scare country could reduce agriculture water scarcity by reducing cropland area for some crop types, and import crop production from other countries.

8. L188ïïjŽ" When $\alpha$ is equal to 1.3, 1.5 and 2.0, the maximum national blue water scarcity in the world is reduced to 6%, 4% and 2%, respectively." In my view, a larger $\alpha$ would result in greater global blue water scarcity reduction, but current study shows the opposite result. So I just wonder the definition of "the maximum national blue water scarcity in the world"?

9. Figure 4. This figure is not clear. Please give the unit and meaning of this figure.

10. Figure 5. There are only tiny differences between figures in the left and right. It's better to show the differences or relative changes.

---

## Author Comment (AC3) · 17 Dec 2019

**Response to comments from reviewer RC2**

Note that reviewer's comments are in italic black, and responses in plain blue font.

*The study on "Changing global cropping patterns to minimize blue water scarcity in the world's hotspots" provides a new view of possibility to reduce crop-related blue water footprint and diminish the severe blue water scarcity worldwide. Plenty work has been done in this study; however, I feel that some parts in the text require careful revisions and improvements before it can be further considered for publication in HESS.*

We appreciate the positive appraisal of the commentator and the useful comments that will be addressed in the following response.

1. *Line 31, you mentioned in the abstract 'changing spatial cropping patterns and international crop trade...", but just showing the 'spatial cropping patterns' changes. It could be much better to look at further on hotspot countries in terms of the responses in trade patterns (just changes in crop trade balances).*

We did consider but decided not to discuss trade balance changes in the paper, to keep the central messages of the paper clear; we agree that the abstract should not suggest otherwise. Discussing changes in international trade patterns will go along with discussing which changes in the cropping pattern would rather increase current trade flows, and which would dampen or reverse current trade flows. The basic underlying message would not be different than in the current manuscript, but the comparison to the reference situation is more complicated than for the cropping pattern.

2. *Line 111, in the introduction of study content, information on how many types of crops considered is lacking.*

The following will be added: "We considered 125 crops of the main crops groups (cereals, fibres, fruits, nuts, oil crops, pulses, roots, spices, stimulants, sugar crops and vegetables) (for an extensive list of crops used see (Chouchane et al., 2019)); optimization was performed using the linear optimization routine from the Optimization Toolbox of MATLAB".

*3. Line 112-113, the first and second constrains seems conflict each other.*

The way how constrains are written now may cause a bit of confusion. A clearer description reads:

"First, **total** rainfed and irrigated harvested areas in each country should not grow beyond their extent in the reference period 1996-2005. Second, the harvested area per country per crop can only expand by a limited rate (which will be varied)".

We thank the reviewer for spotting that and we added the word "total" in the two lines he referred to clearly make the difference between total harvested areas that should not grow beyond the total available harvested areas in the reference period and per crop per country harvested area that could be extended which may result in shifts in cropping patterns.

*4. How do you define the 'cropping pattern' ïïj§*

By cropping pattern, we mean the allocation of crops to rainfed and irrigated land to the countries in the world, where both rainfed and irrigated area of each crop in each country is allowed to be expanded up to a modest maximum rate, while respecting the bounds of current total rainfed and total irrigated area per country.

*5. In the analysis, how the green water limits were considered? I am wondering if there are some places with increasing green WFs but have insufficient green water availability?*

This is a relevant question from a sharp observation. Green water limitation is considered implicitly in the study through consideration of rainfed harvested area and irrigated harvested area separately and by considering rainfed land productivity. Furthermore, the alpha factor is separately applied to the rainfed and irrigated land. Increasing rainfed production could also be the result of shifting crops to more productive crops (higher rainfed land productivity). This can implicitly increase green water consumption, even when that increase is limited by the alpha factor and the differences in green water consumption by crops. The relevance of the effect can be estimated directly from the results of the optimization, and will be added in a resubmitted version.

*6. Line 213, for China you show an 4% increase in BWC. It looks tiny for the whole country, but could matter when such increases in BWC happend in a very severe blue water scarce places within the country. At least some discussion regarding this should be in somewhere of the text. In addition, I am also worry about the assumptions of increasing harvested area per crop so that it could resulted in increases in harvested area in each country, or I could be wrong in understanding the first assumption. Given that for example in China, the national policy is controlling not reducing the total crop harvested*

*area to a level with no possibility to increase anymore... The issue is also important for developing countries facing rapid urbanization in land. Maybe better to discuss this in some points.*

We thank the reviewer for his suggestion. We will add the following in the discussion:

"Changing cropping patterns have reduced global blue water footprint by 9%. However, not all countries benefit the same in the optimized set, India and China, for example, will have a slight increase in their blue water consumption by 5% and 4% respectively. This supports the findings of Davis et al. (2017a) who observed that water scarcity persists in many important agricultural areas (the US Midwest, northern India, Australia's Murray-Darling Basin, for example), indicating that extensive crop production in these places prohibits water sustainability, regardless of crop choice (Davis et al.2017a). In big countries such as India and China, a 4 or 5 % increase in their BWC may seem tiny. However, it could have a negative impact if it occurs in very severe regions of these countries."

About the reviewer's second concern in this comment, the harvested area per country is a constraint in our model. The harvested area for a specific crop could extend by 10% but the total harvested area will remain the same, unless the optimization indicates global production is achieved with less area. Countries will increase the harvested area of the crops in which they have a comparative advantage in terms of blue water and land use and decrease the harvested area of the crops in which they have a comparative disadvantage, this should keep total harvested area per country less or equal to the reference period.

The paper does not consider potential crop land expansions (rainfed or irrigated) to produce additional food to fulfil growing demands, neither does it study effects of improved agricultural practices that may relieve pressure on land and water resources. We agree that the discussion issue raised by the author is relevant in general, but want to restrict specific discussion issues to the scope of the paper.

*7. I get confusions when reading the Discussion. It looks too much limitations to get published, too 'optimized' beyond the real. It may be nice to look into the mass of results and pick some countries with results that really meaningful for local national water management. Please carefully consider about how to interprate in the discussion part. Another limitation should be in caution is the issue related to green water availability, scarcity and limits.*

We will shorten the description of the results in the discussion and highlight the most important changes.

*minor comments:*

*1.  Line 61, better to give the full name of WEF, either in the reference list.*

WEF refers to the World Economic Forum. We will add the full name.

2.  *Table 1, the initial sources of harvested areas or productions should be listed as well.*

The initial source of the harvested areas and productions is FAOSTAT (FAO, 2015). This is now added in Table 1.

---

## Author Comment (AC4) · 21 Dec 2019

**Response to comments from reviewer RC3**

Note that reviewer's comments are in italic black, and responses in plain blue font.

General comments:

*The research on "Changing global cropping patterns to minimize blue water scarcity in the world's hotspots" used a linear optimization algorithm to assess how to change global cropping patterns to reduce blue water-scarce hotspots, with the constraints of global production per crop and current cropland areas. Below are my comments and suggestions:*

We thank the reviewer for his critical comments and suggestions.

*1. The linear optimization algorithm is set for an optimal reduction of blue water scarcity by changing global spatial cropping patterns. The algorithm set an upper limit of the expansion in cropland by a certain maximum rate for each crop per country (the factor $\eth\dot{\,}IZij$), and also limit total cropland to the reference extent. However, there is no lower limit of decrease in cropland area, which means cropland area (or crop production) for some crop types would decrease a lot or even disappear (as shown in results part).*

*Why you set an upper limit, but without a lower limit? If you also set both upper and lower limits of changes in cropland for each crop, do the results change?*

The upper limit is set in order to prevent countries to unrestrictedly expand their cropland in crops where they have comparative advantage. The modest allowed changes in cropping areas of individual crops are aimed to avoid implausible expansions of crop production into cropland areas with significantly different rainfed and irrigated land productivity than where the specific crop is produced currently, due to the heterogeneity within a country (e.g. covering different agroecological zones). However, we do allow countries to decrease their cropland freely without setting a lower limit because here the plausible physical validity of the production characteristics is not compromised. In fact, moving from irrigated production to rainfed production as much as possible is directly related to maximizing the reduction of blue water use and thus blue water scarcity which links to the research objective of this paper.

We expect a significant change in the results if we do set a lower limit to the allowed change in cropland for each crop. The changes will be more apparent for the most water-scarce countries. We will implement your suggestion by performing a sensitivity study to adding lower limits to cropland per country, crop and production system. If relevant we will add some discussion and show some results in the supplementary information of the paper.

*2. Blue water scarcity (BWS): BWS is defined as the total blue water footprint divided by the blue water availability in the country. Here blue water footprint only includes agriculture sector, without water footprint for domestic and industrial. Blue water availability is the natural runoff, which follows Hoekstra et al. (2012), right?*

We acknowledge the validity of the point highlighted by the reviewer. Indeed, blue water has other uses than the agricultural sector (e.g. domestic and industrial). However, the share of agriculture consumptive water use is by far the largest, accounting for 92% of water consumption globally (Hoekstra and Mekonnen, 2012) (mentioned in the submitted version of the paper Line 69-70).

We also thank the reviewer for his suggestion to clarify the definitions of the terms used. We, therefore, added the following:

"Blue water footprint (BWF) refers to the volume of consumptive freshwater use for irrigation that comes from surface and groundwater. Blue water availability is taken from FAO (2015) and refers to the total renewable (internal and external resources) which is the long-term average annual flow of rivers (surface water) and groundwater (FAO, 2003).

*3. L145: "A country is considered to be under low, moderate, significant or severe water scarcity when BWS is lower than 20%, in the range 20-30%, in the range 30-40% and larger than 40%, respectively (Hoekstra et al., 2012)". Hoekstra et al (2012) analysed the BWS at basin level and monthly time scale. But this study assesses water scarcity at country level and annual time scale, I think more discussion is needed to illuminate whether the index used here is suitable.*

We fully agree that considering BWS at national and annual resolution may (and will) hide scarcity localised in time and space. This does limit the interpretability of results at the coarse resolution, and we acknowledge that the discussion on the suitability could be more explicit. We also note that FAO has selected the very similar indicator of Water stress (freshwater withdrawal as a proportion of available freshwater resources) at country and region level as indicator 6.4.2 in the SDG framework (UN-Water, 2018). Next, we will add a variation of the current optimization exercise, contributing to assessing the sensitivity of results to the assumed availability of total renewable freshwater at irrigation areas.

*4. L148: why you choose maximum national blue water scarcity in the world as the indicator for optimization?*

Within the framework of the Sustainable Development Goals, SDG 6.4.2 (Level of water stress), is used as an indicator for Goal 6. Ensure availability and sustainable management of water and sanitation for

all; it is defined similar to water scarcity here, also at the resolution of countries, but based on water extractions rather than consumptive water use. Where lowering the water stress level is a goal for each country, from a global equity perspective lowering stress in countries with highest water scarcity is prioritised. This is operationalised by choosing the maximum national water scarcity as an objective function in the optimization.

*5. There are too much results about the changing cropping patterns and comparative advantages. I think the authors could add more explanation on the mechanism behind the changes, especially for some typical countries.*

We thank the reviewer for his suggestion. We will try to reshape our results section and bring some additional explanation on the mechanism behind the changes for some typical countries.

*6. Discussion part: Previous studies have done a lot of works on the impacts of changing cropping patterns, international food trade and better water productivity on water scarcity (as list in introduction part). I think the discussion part should add more about the similarity and difference between the results in this study and previous studies.*

We will highlight our results in the context of previous studies in the discussion part. For instance, we added the following in the discussion part:

"Changing cropping patterns have reduced global blue water footprint by 9%. However, not all countries benefit the same in the optimized set, India and China, for example, will have a slight increase in their blue water consumption by 5% and 4% respectively. This supports the findings of Davis et al. (2017a) who observed that water scarcity persists in many important agricultural areas (the US Midwest, northern India, Australia's Murray-Darling Basin, for example), indicating that extensive crop production in these places prohibits water sustainability, regardless of crop choice (Davis et al.2017a). In big countries such as India and China, a 4 or 5 % increase in their BWC may seem tiny. However, it could have a negative impact if it occurs in very severe regions of these countries."

*7. More discussions should focus on how the results represented in this study could guide global international food trade, as well as cropping patterns to cope with global water scarcity, especially under future climate change and socioeconomic development. For example, blue water scarcity would intensify in the future as reported in previous studies. And following the results in this study, a water-scare country could reduce agriculture water scarcity by reducing cropland area for some crop types, and import crop production from other countries.*

We will add discussion in the direction suggested by the reviewer. This closely links to comment 5, where we agree that the extensive result reporting took away from highlighting main patterns in findings that can feed into discussions on the role of agricultural trade in water scarcity alleviation policy.

*8. L188ïüjŽ" When α is equal to 1.3, 1.5 and 2.0, the maximum national blue water scarcity in the world is reduced to 6%, 4% and 2%, respectively." In my view, a larger α would result in greater global blue water scarcity reduction, but current study shows the opposite result. So I just wonder the definition of "the maximum national blue water scarcity in the world"?*

Indeed, a higher alpha result in a larger water scarcity reduction. We will rephrase to better emphasize that a WS reduction to a maximum water scarcity of 2% (for alpha = 2) is a further-reaching reduction than a reduction to 6% for alpha =1.3, thus avoiding that *reduced to* is interpreted as *reduced by*.

*9. Figure 4. This figure is not clear. Please give the unit and meaning of this figure.*

We thank the reviewer for his suggestion. We edited the title of Figure 4 to include more information about the Figure and make it easy to understand. The title of the Figure is now the following:

"Relative change in production (production in the optimized cropping pattern divided by the production in the reference situation) per country and per crop group for the case of an optimized cropping pattern with α = 1.5 (relative change = 1: no change, relative change < 1: countries production will be reduced and relative change > 1: countries production will be expanded)".

*10. Figure 5. There are only tiny differences between figures in the left and right. It's better to show the differences or relative changes.*

We agree to the comment and will show the differences in absolute terms (one map for each crop groups showing the difference between the reference situation and the optimized set).

References:

FAO: Review of World Water Resources by Country, Water Reports 23, Food and Agriculture Organization of the United Nations, Rome, Italy, 2003.

UN-Water: Progress on Level of Water Stress – Global baseline for SDG indicator 6.4.2. UN-Water, Geneva, Switzerland, 2018.

---

## Editor Comment (EC1) · Gerrit H. de Rooij (Editor) · 8 Jan 2020

Three reviewers have submitted comments that mainly converge on three main points of critique:

- The global study presented in the paper uses as its smallest unit of analysis entire countries without accounting for local variations in climate, soil conditions, or water availability. Especially in very large countries (India, China, U.S.A.), and smaller if they span different climate zones (e.g., sub-Sahelian Western African countries) this poses a problem. The first reviewer even warns that the results of the study therefore may be misleading.

[Figure]

- The discussion of the results should be refocused (the reviews offer several suggestions) and be made more concise.

- Various unclear passages in the text and/or figures.

The responses by the authors indicated that they see realistic possibilities to improve the discussion and improve the text and the figures to address the second and third of these points. The reviewers offered specific, constructive suggestions. These were not lost on the authors, as their response indicates. They also make some suggestions for modifying the optimization in order to address intra-country variations in water availability. This requires some effort, so at this time it is not clear to what degree the limitation of the nation-based analysis can be remedied. I agree with the reviewers that this is a serious point of concern and I sincerely hope that the authors can improve that aspect of the study. Especially the second comment of reviewer 1 is pertinent. In their response, the authors propose to study the effect of not allowing irrigated areas per country to increase, and the effect of capping the total blue water consumption in each country at current levels. I am not sure if that will be sufficient to mitigating the effect of treating the vast land masses of large countries as uniform entities. In paragraph 3 of the response to reviewer 3, the authors state their intention to expand the analysis to the water availability in irrigated areas of individual countries, if I understand their reply correctly. That would be a useful refinement of the optimization procedure.

My reservation regarding the validity of the country-based analysis notwithstanding, I understand that much of the underlying data are only available on a country-by-country basis. However, Reviewer 1 mentions grid-based data on blue water availability and irrigation areas though, so that could offer some relief of this restriction.

Overall, I believe the paper has potential and will probably benefit from a revision according to the lines suggested in the replies by the authors. I hope they will be able to address, to a certain degree, the concerns raised about larger countries. I therefore invite the authors to revise the paper. Because this will involve additional calculations,

I will classify this as a major revision. I am, of course, aware that the author team will have to carry on without the input of Prof. Hoekstra, who unfortunately passed away, and whose input will undoubtedly be sorely missed. I therefore ask the authors to inform me if more time is needed, so I can arrange an extension of the deadline if so desired.

---

## Author Response (AR1)

We are thankful to have the opportunity to resubmit our revised paper. The reviewer comments were very helpful to clarify and improve the original manuscript. All comments by the reviewers have been addressed, and corresponding changes have been made in the manuscript where necessary. Below, a detailed point-wise response to the reviewer's remarks and marked-up manuscript version.

Note that reviewer's comments are in italic black, and responses in plain blue font.

General comments:

**Response to comments from reviewer RC1**

*The authors determine for a large number of crops how crop production could be shifted among the countries of the world to produce the same amount of each crop globally while minimizing the highest value of a country-scale indicator of blue water scarcity, without any extension of the total national cropland but a with a certain maximum allowed extension of cropping area in the countries, both for rainfed and irrigated production. Mainly for reasons described as limitations of the study by the authors themselves (lines 368-378 but also 379-383) I think that the results of the study are not informative and even misleading. This is due to the scale of the study which inclusively considers countries as homogeneous units of analysis, regarding land and water productivities as well as blue water availability.*

We thank the reviewer for his critical comments. As the reviewer already noted, most limitations observed in the comments are acknowledged and described in the paper's discussion. We took a number of actions in order to soften most of the study limitations (by limiting areal crops expansion to a maximum factor-alpha for instance). The main issue here thus is the extent to which the usage of country-average data and the interpretation of results is appropriate. Firstly, one relevant methodological aspect appears misinterpreted: the allowed land-use changes at country level (limited by factor-alpha) is not an allowable expansion in rainfed / irrigated crop area per country limited by national agricultural area, but rather is an allowable shift in the cropping pattern within the bounds of current rainfed and irrigated area per country. So current production characteristics on currently irrigated lands are not assumed to be valid elsewhere. The modest allowed changes in cropping areas of individual crops prevent significant shifts in crop allocation within a country (e.g. to other agro-ecological zones), avoiding implausible results due to the heterogeneity in rainfed and irrigated land productivity. The impact of the observed heterogeneity in blue water availability can be more influential. Water availability is a complex variable because the same volume of water at a specific location and time can be considered available for use at any downstream location and (if storages are present) at any moment in the year; countries base their water management on these properties and implement policies of water allocation within a river basin, reservoir construction and management and large scale inter-basin water transfers. The extent of such policies to justify only considering total national freshwater availability in assessing water scarcity is limited, however, calling for care in interpreting national-scale water scarcity as an indicator and in performing scenario exercises as in the present manuscript. This discussion was underemphasized in the original manuscript but is now explicitly addressed in the paper. This discussion closely links to considerations on the choice of *Water stress* (freshwater withdrawal as a proportion of available freshwater resources) at country and region level as indicator 6.4.2 in the SDG framework (FAO, 2018).

We added the following:

In the introduction part: "The spatial resolution of the country level reflects the coarse resolution at which FAO monitors and reports water stress in the SDG framework (FAO, 2018); subnational heterogeneity in water scarcity, that is significant in countries like USA or China, is not covered at this resolution". **(Line 120-122)**

In the discussion part: "We minimize average water scarcity in countries; within countries scarcity differences will still appear, both in the reference situation and in the case of the optimized cropping patterns. Still, water scarcity indicators at national levels provide insight; within the framework of the Sustainable Development Goals, indicator 6.4.2 (Level of water stress), is used to monitor Goal 6 (Ensure availability and sustainable management of water and sanitation for all); it is defined similar to water scarcity in our study, also at the resolution of countries, but based on water extractions rather than consumptive water use. Where lowering the water stress level is a goal for each country, from a global equity perspective lowering stress in countries with highest water scarcity is prioritised. This is operationalised by choosing the maximum national water scarcity as an objective function in the optimization. Relieving water scarcity in specific hotspots within countries by changing cropping patterns could be studied using the current approach but is beyond the scope of this paper. The sensitivity analysis did show that by far the largest impact on water scarcity relief emerges from shifts in cropping patterns of rainfed crops, not depending on the heterogeneity of blue water availability; therefore water scarcity reduction in countries with highest scarcity at national level in the current study does not rely on worsening water scarcity in countries with heterogeneous conditions". **(Line 412-424)**

Next, we added a variation of the current optimization exercise, contributing to assessing the sensitivity of results to the assumed availability of total renewable freshwater at irrigation areas (see response to the next comment).

It should be noted here, that by far the largest impact on water scarcity relief emerges from shifts in cropping patterns of rainfed crops, not depending on the heterogeneity of blue water availability as shown in the sensitivity analysis added to the paper results. The dominance of this aspect of the changed global cropping pattern is illustrated using an additional optimization exercise to separate out this effect (see response to the next comment).

Where the scale of analysis chosen in the paper calls for careful introduction of the definition of the exercise and its interpretation, to our knowledge it considers, for the first time, both differences in water productivities and in water endowments to analyse comparative advantages of countries for different types of crop production.

*The novelty claimed in the manuscript is consideration of blue water scarcity. Unfortunately, blue water scarcity is only considered as one value per country, computed as the ratio of total blue water use in the the country and blue water availability in the country. This is problematic as their are important crop-producing large countries like India, China and the US (but also Australia) with humid and semi-arid climate zone, where irrigated crop production and thus blue water use is concentrated in the semi-arid/arid regions of the country while blue water availability is high the humid parts of the country. This is why these countries, in which large regions suffer from irrigation-induced water stress and even groundwater depletion, do not appear among the 21 countries with the highest water scarcity (Table 2) for which the authors show to what extent blue water consumption and thus blues water scarcity could be reduced by shifting crop production to other countries (with lower blue water scarcity). One result is that in the optimized distribution of crop production among countries, both China, India and Australia increase their blue water consumption (Fig. 2 bottom). I do not find it plausible that the thus optimized distribution of crop production among countries "minimizes blue water scarcity in the worlds's hotspots"(as is formulated in the title).*

*I think it is a prerequisite for publication of the study that the authors show the results of a sensitivity analysis regarding the spatial analysis units. Blue water availability values as well as irrigated areas are available at a spatial resolution of 0.5◦ by 0.5◦ , and this information could be used to see how the optimization results change if the blue water availability in the irrigated areas/cropping areas are taken into account instead of average country values. You could have a look at Yano et al. 2015 (Yano S, Hanasaki N, Itsubo N and Oki T 2015 Water scarcity footprints by considering the differences in water sources Sustainability 7 9753–72) where water scarcity at the country and for irrigated areas are computed separately and compared. Blue water availability from various global hydrological model available at www.isimip.com could be used.*

The above comment closely connects to the first one. We agree that the term 'hotspot', meant to indicate the world's most water-scarce countries, can easily be misinterpreted. Therefore, we removed the term from the title, and only use the term in the body of the text in discussing limitations to interpretability the results at national scale due to heterogeneity. We also agree that modest to low water scarcity indicators at national level may hide hotspots within a country; we do note however that still water stress or water scarcity are widely used as indicators for the human pressure on water resources as national scale, e.g. SDG 6.4.2 Water Stress in FAO's AQUASTAT, intended for country comparisons in global studies.

The optimization has been updated. While the objective function and most constraints remain the same, we now disallow increases in blue water use in each country. All results have been updated accordingly. Moreover, in order to identify the impact of restricting expansion to rainfed areas only, a sensitivity analysis has been conducted showing the share of effects of shifts in rainfed areas only in the total effects when allowing both rainfed and irrigated areas to increase by the factor α. The sensitivity results show the dominance of only shifting crops within the rainfed area in the contribution of reducing maximum blue water scarcity.

*In addition, it is necessary to broaden the literature review. For example, the work of Taikan Oki and his group have not been considered. Please review Oki et al. 2017, Environ. Res. Lett. 12 044002 and some of the references therein. Oki and Kanae 2004 already showed global water savings by global trade.*

A number of relevant citations, including the ones suggested by the reviewer, has been added to the paper's introduction.

*Specific comments*

*L76: Jalava et al. 2016 also studied the effect of food loss reduction (https://doi.org/10.1002/2015EF000327)*

This citation and another relevant one has been added to highlight the effect of food loss reduction on water use.

*L79: Explain more clearly to a broader audience what the definition of virtual water is (also: does not only relate to food).*

The definition of virtual water has been changed into the following: The trade in 'embedded water' ( also known as virtual water trade) is the hidden flow of water if food or other commodities are traded from one place to another (Allan, 1998). **(Line 78-80)**

*L102ff. Explain more clearly the study of Davis et al. 2017a, and compare their methods and results to your study (e.g. in the discussion section).*

We add the following:

In the introduction: "However, the current study has a number of differences with Davis et al., (20017a). First, we are only changing cropping patterns while maintaining the same global production per crop whereas Davis et al. (2017a) aim for a higher caloric and protein production while reducing water use; that also results in a different global consumption pattern, which hampers the identification of potential water saving effects of just production shifts amongst countries. Second, we consider a larger number of crops (125 crops including vegetables, fruits and pulses which were not considered in Davis et.al., (2017a) study)." **(Line 109-114)**

In the discussion, we add: "The current study supports the findings of Davis et al., (2017a) on the benefits of crop redistribution on water saving which could be a potential strategy for sustainable crop production and an alternative to the large investments that are usually needed to close up the technological and yield gaps in developing nations." **(Line 440-443)**

"Changing cropping patterns could reduce global blue water footprint by 21% and global irrigated area by 10%. These findings prove that current high scarcity levels in a serious number of countries is shown to be caused by the current crop allocation pattern, rather than by an inevitability of those scarcities to occur; that suggests that water endowment is insufficiently driving crop allocation to avoid water scarcity. This in consistent with Zhao et al., (2019) who find in their study for China that comparative advantages with respect to labour and water were not reflected in the regional distribution of agricultural production. However, not all countries would benefit similarly in the optimized set, India and China, main crop producers in the reference situation, will only start to have a decrease in their blue water scarcity when the allowed expansion rate is larger than 20%. This is in line with the findings of Davis et al., (2017a) who find in their simulations that water scarcity persists in many important agricultural areas (the US Midwest, northern India, Australia's Murray-Darling Basin, for example), indicating that extensive crop production in these places prohibits water sustainability, regardless of crop choice (Davis et al., 2017a)." **(Line 453-463)**

*L111: Define clearly here that "cropping patterns "mean the distribution of production of a certain crop among the nations/countries but not within.*

We add the following explanation: "(With cropping pattern we mean the allocation of crops to rainfed and irrigated land in all countries in the world, where both rainfed and irrigated area of each crop in each country is allowed to expand up to a modest maximum rate (factor α), while respecting the bounds of current total rainfed and total irrigated area per country as well as the global production per crop.)". **(Line 122-125)**

*L118: Expand methods section with respect to considered crops/crop groups, algorithm for optimization, e.g. how was ensemble of potential cropping patterns produced?*

We add: "We considered 125 crops of the main crops groups (cereals, fibres, fruits, nuts, oil crops, pulses, roots, spices, stimulants, sugar crops and vegetables; for an extensive list of crops used see (Chouchane et al., 2019)); optimization was performed using the linear optimization routine from the Optimization Toolbox of MATLAB". **(Line 147-149)**

*L139: BWS only takes into account irrigation water use but not the other use sectors. Define blue water footprint.*

We added the following: "Blue water footprint (BWF) refers to the volume of consumptive freshwater use for irrigation that comes from surface and groundwater". **(Line 160-161)**

*L159: Explain why you chose to minimize (only) the highest national blue water scarcity.*

We minimize average water scarcity in countries; within countries scarcity differences will still appear, both in the reference situation and in the case of the optimized cropping patterns. Still, water scarcity indicators at national levels provide insight; within the framework of the Sustainable Development Goals, indicator 6.4.2 (Level of water stress), is used to monitor Goal 6 (Ensure availability and sustainable management of water and sanitation for all); it is defined similar to water scarcity in our study, also at the resolution of countries, but based on water extractions rather than consumptive water use. Where lowering the water stress level is a goal for each country, from a global equity perspective lowering stress in countries with highest water scarcity is prioritised. This is operationalised by choosing the maximum national water scarcity as an objective function in the optimization. Relieving water scarcity in specific hotspots within countries by changing cropping patterns could be studied using the current approach but is beyond the scope of this paper. This has been added to the paper's discussion. **(Line 412-421)**

*L220-364. Please shorten the lengthy description of the changing cropping patterns and comparative advantages shown in figures and tables but try to explain the results.*

This has been shortened and the results section has been reshaped.

*L367ff Also discuss the real-life meaning and consequences of optimized global cropping pattern, in particular reduced blue water consumption in the countries listed in Table 2. E.g. if BWC is reduced*

*from 1900 to 280 million m3/yr in Libya, crop production (Fig. 4) and income would be strongly reduced, too. Could the production/income loss be somehow related to GDP to understand the problems that would result from the analyzed global-scale optimization?*

Consequences of the changes in the global cropping pattern on agricultural economy, farm economy and food self-sufficiency are outside of the scope of this paper. Changes towards the optimized cropping patterns identified here would require agroeconomic policies, e.g. on commodity prices, price- and farm income subsidies or trade regulations to reflect implicit resource use.

We already mentioned some impacts related to reduced production in real life. We mentioned the countries with the largest decrease in their blue water footprint of crop production (last paragraph in the discussion) and the impact that could result from that. However, this doesn't mean directly that the total production is reduced. Since for some countries, when possible, they will switch to rainfed production. So, income reduction is not necessarily proportional to the reduction in blue water consumption. To be able to assess the impact of the reduction in BWC on the country GDP we should be able to trace back the consumption per crop per country and initial import and export. By calculating the changes in consumption, import and export we could assess the changes in the GDP. This is out of the paper scope for now.

*L408 ff. I would not use the grammatic form of "will", e.g. in "Cereal production will get reduced in Africa". Maybe better: "If blue water scarcity was globally optimized, cereal production would be reduced in Africa according to our analysis."*

The paper has been improved textually.

**Response to comments from reviewer RC2**

Note that reviewer's comments are in italic black, and responses in plain blue font.

*The study on "Changing global cropping patterns to minimize blue water scarcity in the world's hotspots" provides a new view of possibility to reduce crop-related blue water footprint and diminish the severe blue water scarcity worldwide. Plenty work has been done in this study; however, I feel that some parts in the text require careful revisions and improvements before it can be further considered for publication in HESS.*

We appreciate the positive appraisal of the commentator and the useful comments that will be addressed in the following response.

1.  *Line 31, you mentioned in the abstract 'changing spatial cropping patterns and international crop trade...", but just showing the 'spatial cropping patterns' changes. It could be much better to look at further on hotspot countries in terms of the responses in trade patterns (just changes in crop trade balances).*

We did consider but decided not to discuss trade balance changes in the paper, to keep the central messages of the paper clear; we agree that the abstract should not suggest otherwise. Discussing changes in international trade patterns will go along with discussing which changes in the cropping pattern would rather increase current trade flows, and which would dampen or reverse current trade flows. The basic underlying message would not be different than in the current manuscript, but the comparison to the reference situation is more complicated than for the cropping pattern.

2.  *Line 111, in the introduction of study content, information on how many types of crops considered is lacking.*

The following has been added: "We considered 125 crops of the main crops groups (cereals, fibres, fruits, nuts, oil crops, pulses, roots, spices, stimulants, sugar crops and vegetables; for an extensive list of crops used see (Chouchane et al., 2019)); optimization was performed using the linear optimization routine from the Optimization Toolbox of MATLAB". **(Line 147-149)**

3.  *Line 112-113, the first and second constrains seems conflict each other.*

The way how constrains are written now may cause a bit of confusion. A clearer description reads:

"First, **total** rainfed and irrigated harvested areas in each country should not grow beyond their extent in the reference period 1996-2005. Second, the harvested area per country per crop can only expand by a limited rate (which will be varied), both for the rainfed and irrigated area." **(Line 126-129)**

We thank the reviewer for spotting that and we added the word "total" in the two lines he referred to clearly make the difference between total harvested areas that should not grow beyond the total available harvested areas in the reference period and per crop per country harvested area that could be extended which may result in shifts in cropping patterns.

4. *How do you define the 'cropping pattern'.*

The following explanation has been added to the paper's introduction: With cropping pattern we mean the allocation of crops to rainfed and irrigated land in all countries in the world, where both rainfed and irrigated area of each crop in each country is allowed to expand up to a modest maximum rate (factor α), while respecting the bounds of current total rainfed and total irrigated area per country as well as the global production per crop. **(Line 122-125)**

*5. In the analysis, how the green water limits were considered? I am wondering if there are some places with increasing green WFs but have insufficient green water availability?*

This is a relevant question from a sharp observation. Green water limitation is considered implicitly in the study through consideration of rainfed harvested area and irrigated harvested area separately and by considering rainfed land productivity. Furthermore, the alpha factor is separately applied to the rainfed and irrigated land. Increasing rainfed production could also be the result of shifting crops to more productive crops (higher rainfed land productivity). This can implicitly increase green water consumption, even when that increase is limited by the alpha factor and the differences in green water consumption by crops. The relevance of the effect is estimated in the sensitivity analysis added to the updated version of the paper.

*6. Line 213, for China you show an 4% increase in BWC. It looks tiny for the whole country, but could matter when such increases in BWC happend in a very severe blue water scarce places within the country. At least some discussion regarding this should be in somewhere of the text. In addition, I am also worry about the assumptions of increasing harvested area per crop so that it could resulted in increases in harvested area in each country, or I could be wrong in understanding the first assumption. Given that for example in China, the national policy is controlling not reducing the total crop harvested area to a level with no possibility to increase anymore... The issue is also important for developing countries facing rapid urbanization in land. Maybe better to discuss this in some points.*

We thank the reviewer for his suggestion. The optimization algorithm has been updated in a way that an increase in blue water use is no longer allowed. The optimization will try to reduce the water scarcity of all countries starting by the most water-scarce countries when the allowed expansion is low.

About the reviewer's second concern in this comment, the harvested area per country is a constraint in our model. The harvested area for a specific crop could extend by 10% but the total harvested area will remain the same, unless the optimization indicates global production is achieved with less area. Countries will increase the harvested area of the crops in which they have a comparative advantage in terms of blue water and land use and decrease the harvested area of the crops in which they have a comparative disadvantage, this should keep total harvested area per country less or equal to the reference period.

The paper does not consider potential crop land expansions (rainfed or irrigated) to produce additional food to fulfil growing demands, neither does it study effects of improved agricultural practices that may relieve pressure on land and water resources. We agree that the discussion issue raised by the author is relevant in general, but want to restrict specific discussion issues to the scope of the paper.

*7. I get confusions when reading the Discussion. It looks too much limitations to get published, too 'optimized' beyond the real. It may be nice to look into the mass of results and pick some countries with results that really meaningful for local national water management. Please carefully consider about how to interprate in the discussion part. Another limitation should be in caution is the issue related to green water availability, scarcity and limits.*

The description of the results has been shortened and most important changes has been highlighted and discussed in the discussion

*minor comments:*

*1. Line 61, better to give the full name of WEF, either in the reference list.*

WEF refers to the World Economic Forum. The full name is specified in the reference list.

*2. Table 1, the initial sources of harvested areas or productions should be listed as well.*

The initial source of the harvested areas and productions is FAOSTAT (FAO, 2015). This is now added in Table 1.

**Response to comments from reviewer RC3**

Note that reviewer's comments are in italic black, and responses in plain blue font.

General comments:

*The research on "Changing global cropping patterns to minimize blue water scarcity in the world's hotspots" used a linear optimization algorithm to assess how to change global cropping patterns to reduce blue water-scarce hotspots, with the constraints of global production per crop and current cropland areas. Below are my comments and suggestions:*

We thank the reviewer for his critical comments and suggestions.

*1. The linear optimization algorithm is set for an optimal reduction of blue water scarcity by changing global spatial cropping patterns. The algorithm set an upper limit of the expansion in cropland by a certain maximum rate for each crop per country (the factor ðˈIZij), and also limit total cropland to the reference extent. However, there is no lower limit of decrease in cropland area, which means cropland area (or crop production) for some crop types would decrease a lot or even disappear (as shown in results part).*

*Why you set an upper limit, but without a lower limit? If you also set both upper and lower limits of changes in cropland for each crop, do the results change?*

The upper limit is set in order to prevent countries to unrestrictedly expand their cropland in crops where they have comparative advantage. The modest allowed changes in cropping areas of individual crops are aimed to avoid implausible expansions of crop production into cropland areas with significantly different rainfed and irrigated land productivity than where the specific crop is produced currently, due to the heterogeneity within a country (e.g. covering different agroecological zones). However, we do allow countries to decrease their cropland freely without setting a lower limit because here the plausible physical validity of the production characteristics is not compromised. In fact, moving from irrigated production to rainfed production as much as possible is directly related to maximizing the reduction of blue water use and thus blue water scarcity which links to the research objective of this paper.

Explicitly setting a lower limit to the allowed change in cropland for each crop will obviously have a significant change in the results. The changes will be more apparent for the most water-scarce countries. If for example we enforce countries to reduce production per crop and production system by at most 50%, the water scarcity will remain at least 50% of the actual one. We added a discussion about the trade-off between the global objective of countries jointly reducing the global blue water scarcity and about the effect of that on each individual country, for example the increase of food import dependency for some countries. we decided not to add alternative formal optimizations to further substantiate this discussion point as results are very predictable and does not significantly contribute to the current paper's objective.

*2. Blue water scarcity (BWS): BWS is defined as the total blue water footprint divided by the blue water availability in the country. Here blue water footprint only includes agriculture sector, without water footprint for domestic and industrial. Blue water availability is the natural runoff, which follows Hoekstra et al. (2012), right?*

We acknowledge the validity of the point highlighted by the reviewer. Indeed, blue water has other uses than the agricultural sector (e.g. domestic and industrial). However, the share of agriculture consumptive water use is by far the largest, accounting for 92% of water consumption globally (Hoekstra and Mekonnen, 2012) (mentioned in the submitted version of the paper Line 69-70).

We also thank the reviewer for his suggestion to clarify the definitions of the terms used. We, therefore, added the following:

"Blue water scarcity (BWS) is defined per country i as the total blue water footprint divided by the blue water availability in the country (Hoekstra et al., 2012). The blue water footprint (BWF) refers to the volume of consumptive freshwater use for irrigation that comes from surface and groundwater. Blue water availability is taken from FAO (2015) and refers to the total renewable (internal and external resources) which is the long-term average annual flow of rivers (surface water) and sustainably available groundwater (FAO, 2003)". **(Line 159-163)**

*3. L145: "A country is considered to be under low, moderate, significant or severe water scarcity when BWS is lower than 20%, in the range 20-30%, in the range 30-40% and larger than 40%, respectively (Hoekstra et al., 2012)". Hoekstra et al (2012) analysed the BWS at basin level and monthly time scale. But this study assesses water scarcity at country level and annual time scale, I think more discussion is needed to illuminate whether the index used here is suitable.*

We fully agree that considering BWS at national and annual resolution may (and will) hide scarcity localised in time and space. This does limit the interpretability of results at the coarse resolution, and we acknowledge that the discussion on the suitability could be more explicit. We also note that FAO has selected the very similar indicator of Water stress (freshwater withdrawal as a proportion of available freshwater resources) at country and region level as indicator 6.4.2 in the SDG framework (UN-Water, 2018). Next, we added a variation of the current optimization exercise, contributing to assessing the sensitivity of results to the assumed availability of total renewable freshwater at irrigation areas. The sensitivity analysis showed that the shifts in rainfed areas only had a dominant share in reducing the maximum blue water scarcity for different expansion factors α, as is discussed in the paper.

*4. L148: why you choose maximum national blue water scarcity in the world as the indicator for optimization?*

We minimize average water scarcity in countries; within countries scarcity differences will still appear, both in the reference situation and in the case of the optimized cropping patterns. Still, water scarcity indicators at national levels provide insight; within the framework of the Sustainable Development Goals, indicator 6.4.2 (Level of water stress), is used to monitor Goal 6 (Ensure availability and sustainable management of water and sanitation for all); it is defined similar to water scarcity in our study, also at the resolution of countries, but based on water extractions rather than consumptive water use. Where lowering the water stress level is a goal for each country, from a global equity perspective lowering stress in countries with highest water scarcity is prioritised. This is operationalised by choosing the maximum national water scarcity as an objective function in the optimization. Relieving water scarcity in specific hotspots within countries by changing cropping patterns could be studied using the current approach but is beyond the scope of this paper. This has been added to the paper's discussion. **(Line 412-421)**

*5. There are too much results about the changing cropping patterns and comparative advantages. I think the authors could add more explanation on the mechanism behind the changes, especially for some typical countries.*

We thank the reviewer for his suggestion. The results section has been reshaped and some main finding of typical countries has been highlighted. We also added some discussion of major crops producers' countries in the paper's discussion part:

"Findings suggest that China, one of the main producers of the major crop in the world, will abandon soybean production and halve wheat irrigation area. This will relieve some of the pressure on the northern part of China where water scarcity is the most severe (Ma et al., 2020). China will increase the harvested area of rice and rapeseed, the crops with the most significant comparative advantage in terms of land and water. Similarly, our results suggest that the US, another major crops producer, would and restrict soybean production to rainfed systems, abandoning irrigation, in the optimized set in the US. The US focuses on producing maize, mainly rainfed, for which the US has a comparative advantage in terms of water and land productivities. This may be a great relief to the US corn belt where most of irrigated soybeans and maize are located (Zhong et al., 2016) and could be a remedy to the projected water shortage of that region resulting from population growth and climate change (Brown et al., 2019). We also find that India, another major producer of crops in the world, will move away from sorghum production and shift a large share of its rice and wheat production to rainfed conditions. Moving to rainfed production in India could mitigate the effect of the intensive use of irrigation from groundwater and surface water which caused groundwater degradation in many districts of Haryana and Punjab, the largest contributing states to rice and wheat production in India (Singh, 2000)". **(Line 464-476)**

*6. Discussion part: Previous studies have done a lot of works on the impacts of changing cropping patterns, international food trade and better water productivity on water scarcity (as list in introduction part). I think the discussion part should add more about the similarity and difference between the results in this study and previous studies.*

We highlighted our results in the context of previous studies in the discussion part. For instance, we added the following:

"The current study supports the findings of Davis et al., (2017a) on the benefits of crop redistribution on water saving which could be a potential strategy for sustainable crop production and an alternative to the large investments that are usually needed to close up the technological and yield gaps in developing nations. Besides reducing water and land use, changing cropping pattern will also have an impact on reducing GHG emission that results from extensive energy activities in irrigation such as groundwater pumping which accounted for around 61% of total irrigation emissions in China (Zou et al., 2015)". **(Line 440-444)**

"Changing cropping patterns could reduce global blue water footprint by 21% and global irrigated area by 10%. These findings prove that current high scarcity levels in a serious number of countries is shown to be caused by the current crop allocation pattern, rather than by an inevitability of those scarcities to occur; that suggests that water endowment is insufficiently driving crop allocation to avoid water scarcity. This in consistent with Zhao et al., (2019) who find in their study for China that comparative advantages with respect to labour and water were not reflected in the regional distribution of agricultural production. However, not all countries would benefit similarly in the optimized set, India and China, main crop producers in the reference situation, will only start to have a decrease in their blue water scarcity when the allowed expansion rate is larger than 20%. This is in line with the findings of Davis et al., (2017a) who find in their simulations that water scarcity persists in many important agricultural areas (the US Midwest, northern India, Australia's Murray-Darling Basin, for example), indicating that extensive crop production in these places prohibits water sustainability, regardless of crop choice (Davis et al., 2017a)". **(Line 453-463)**

*7. More discussions should focus on how the results represented in this study could guide global international food trade, as well as cropping patterns to cope with global water scarcity, especially under future climate change and socioeconomic development. For example, blue water scarcity would intensify in the future as reported in previous studies. And following the results in this study, a water-scare country could reduce agriculture water scarcity by reducing cropland area for some crop types, and import crop production from other countries.*

We added discussion in the direction suggested by the reviewer. This closely links to comment 5, where we agree that the extensive result reporting took away from highlighting main patterns in findings that can feed into discussions on the role of agricultural trade in water scarcity alleviation policy.

*8. When α is equal to 1.3, 1.5 and 2.0, the maximum national blue water scarcity in the world is reduced to 6%, 4% and 2%, respectively. " In my view, a larger α would result in greater global blue water scarcity reduction, but current study shows the opposite result. So, I just wonder the definition of "the maximum national blue water scarcity in the world"?*

Indeed, a higher alpha result in a larger water scarcity reduction. The sentence has been rephrased to better emphasize that a WS reduction to a maximum water scarcity of 2% (for alpha = 2) is a further-reaching reduction than a reduction to 6% for alpha =1.3, thus avoiding that *reduced to* is interpreted as *reduced by*.

*9. Figure 4. This figure is not clear. Please give the unit and meaning of this figure.*

We thank the reviewer for his suggestion. We edited the title of Figure 4 to include more information about the Figure and make it easy to understand. The title of the Figure is now the following:

"Absolute change in production for cereals, fruits, oil crops, sugar crops and vegetables per country (in $10^6$ t/yr) (maps on the left hand) and relative production (ratio of production in optimized and reference situation) for the same crops groups for the case of an optimized cropping pattern with α=1.5 (maps on the right hand), all compared to the reference cropping period (1996-2005): relative production = 1: no change, relative production < 1: countries production is reduced and relative production > 1: countries production is expanded".

*10. Figure 5. There are only tiny differences between figures in the left and right. It's better to show the differences or relative changes.*

We agree to the comment and we changed both Figures 4 and 5. The new figures show both absolute and relative changes in production for all considered crop groups.

[revised manuscript text omitted]

Most countries with severe water scarcity (BWS>40%) in the reference situation will haveshow a moderate (BWS in the range 20-30%) to low water scarcity (BWS<20%) in the optimized situation with α = 1.1 (Figure 1). The blue water scarcity reduction in most countries comes at the price of a slight increase in BWS of some countries. In India, BWS increases from

However, not all countries would benefit similarly in the optimized situation. China and India, major crops producers in the reference situation, only start to have a decrease in their BWS when α ≥ 1.3.

[Figure]

[Figure]

**Figure 1.** Current and optimized (α = 1.1) blue water scarcity.

In the case of α = 1.1, Pakistan, the 3rd largest consumer of blue water in the reference situation, has the largest reduction in its blue water consumption in absolute terms, viz. 60,000 m³/yr, which represents 80% of its current BWC and 35% of the global blue water saving. Other countries that have a significant reduction in their BWC in absolute terms include Iran, Egypt, Iraq, Syria, Saudi Arabia, Sudan and Turkmenistan  (Figure 2). However, not all countries  would benefit similarly in the optimized set, India and China, the first and second largest consumer of blue water in reference situation, will only start to have a  decrease in their blue water scarcity when the allowed expansion rate α is larger than 1.2; this is due to the optimization of water scarcity at the level of countries, where India and China have modest national water scarcity.

[Figure]

[Figure]

**Figure 2.** Current blue water consumption (BWC)depth in mm/yr and blue water saving as a percentage of current BWC in the case of an optimized cropping pattern ($\alpha$ = 1.1).

**The changing global cropping pattern for the case of $\alpha$ = 1.1**

The reduction of global blue water consumption is achieved by reallocating the most resource-intensive crops from countries that initially have a high BWS to countries that have a have lower BWS and higher productivity in terms of land and water. Cereal to countries with significantly higher productivities, both for rainfed and irrigated production will be, and thus reducing irrigation in countries that initially have a high BWS. In the optimised cropping pattern, cereal production is reduced most significantly in Africa and the Americas, relative to the reference situation, and South America and expanded in North

America and Europe and Asia (Table 3). Irrigated cereal production will beis reduced in allmost world regions (except for a small expansion in Europe and South America) whereas global rainfed production increases. In Africa, For individual countries, Pakistan and Egypt will haveis the biggest percentage oflargest decrease in total cereal production decrease. The .

[revised manuscript text omitted]